# Equi-mRNA: Protein Translation Equivariant Encoding for mRNA Language Models

**Mehdi Yazdani-Jahromi**
Department of Computer Science
University of Central Florida
Orlando, FL 32816
yazdani@ucf.edu

**Ali Khodabandeh Yalabadi**
Department of Industrial Engineering
University of Central Florida
Orlando, FL 32816
yalabadi@ucf.edu

**Ozlem Ozmen Garibay**
Department of Computer Science and Industrial Engineering
University of Central Florida
Orlando, FL 32816
ozlem@ucf.edu

## Abstract

The growing importance of mRNA therapeutics and synthetic biology highlights the need for models that capture the latent structure of synonymous codon (different triplets encoding the same amino acid) usage, which subtly modulates translation efficiency and gene expression. While recent efforts incorporate codon-level inductive biases through auxiliary objectives, they often fall short of explicitly modeling the structured relationships that arise from the genetic code's inherent symmetries. We introduce Equi-mRNA, the first codon-level equivariant mRNA language model that explicitly encodes synonymous codon symmetries as cyclic subgroups of 2D Special Orthogonal matrix ($SO(2)$). By combining group-theoretic priors with an auxiliary equivariance loss and symmetry-aware pooling, Equi-mRNA learns biologically grounded representations that outperform vanilla baselines across multiple axes. On downstream property-prediction tasks including expression, stability, and riboswitch switching Equi-mRNA delivers up to $\approx 10\%$ improvements in accuracy. In sequence generation, it produces mRNA constructs that are up to $\approx 4\times$ more realistic under Fréchet BioDistance metrics and $\approx 28\%$ better preserve functional properties compared to vanilla baseline. Interpretability analyses further reveal that learned codon-rotation distributions recapitulate known GC-content biases and tRNA abundance patterns, offering novel insights into codon usage. Equi-mRNA establishes a new biologically principled paradigm for mRNA modeling.

## 1 Introduction

RNA analysis has become central to modern molecular biology due to RNA's essential regulatory and functional roles within cellular systems [25, 13]. Among RNA molecules, messenger RNA (mRNA) is especially critical, serving as a direct translator of genetic information into functional proteins, thereby underpinning both fundamental biological processes and therapeutic advancements [40]. At the nucleotide level, mRNA sequences are structured as triplets known as codons, each specifying a single amino acid within the resultant protein. Due to redundancy in the genetic code, multiple synonymous codons can encode the same amino acid, a surjective many-to-one relationship that introduces substantial complexity. Importantly, synonymous codons are not utilized equally; This phenomenon, termed codon bias, varies widely across species and even between genes within the

same organism, influenced by both mutational biases and selective pressures aimed at optimizing translational efficiency [35]. Thus, synonymous codons, despite coding for identical amino acids, can differentially impact mRNA stability, translational speed, and protein folding dynamics [26, 51, 5]. Such subtle yet critical differences have profound biological implications, with numerous synonymous mutations implicated in human diseases [31].

Understanding and leveraging codon-level nuances is therefore essential, particularly for applications such as precision-engineered mRNA vaccines, gene therapies, and synthetic biology tools [6, 33].

Recent advances have demonstrated the significant potential of language models in analyzing biological sequences, particularly for proteins [11, 12, 24, 16] and DNA [28, 56]. For messenger RNA, this success has not held, in part because biological sequences follow constraints and grammars that are inherently different from natural language.

Given the rich biological signal embedded in codon usage, there is strong motivation to develop specialized language models tailored explicitly for coding regions of the genome. Such models could exploit inductive biases inherent in the genetic code such as the codon hierarchy [52] and organism-specific codon preferences to significantly enhance predictive modeling of biologically relevant properties.

To fully realize the potential of language models in mRNA modeling, we developed a theoretically principled framework for representing mRNA codon sequences, explicitly incorporating the biological symmetry and redundancy inherent in the genetic code (Fig. 1b). Using orthogonal group actions with equivariance constraints, implemented as per-subspace rotations of shared amino acid embeddings, we produce interpretable embeddings that encode synonymous codon relationships. Integrating these biologically-informed inductive biases significantly improve the data efficiency, robustness, generalizability, and generative performance of RNA-focused language models.

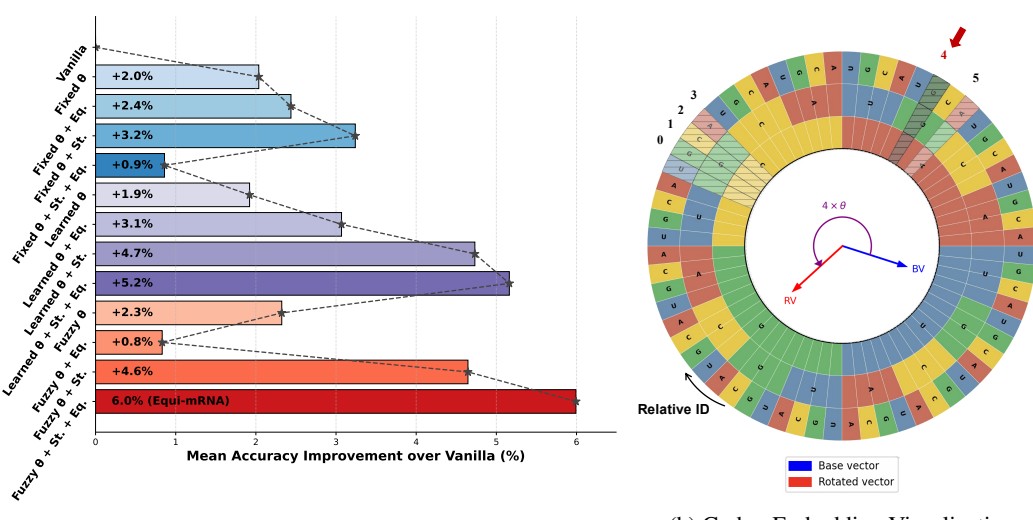

(a) Component impact

(b) Codon Embedding Visualization

Figure 1: Equi-mRNA Embedding Visualization: (a) Mean accuracy improvement over vanilla codon embeddings (Plain learned codon embeddings) for all twelve SO(2) variants: fixed, learned, and fuzzy rotation generators, each with and without equivariance. (b) Codon wheel for arginine: inner, middle, and outer rings denote the 1st, 2nd, and 3rd nucleotide positions, and codons are ordered by relative ID. A shared base vector (BV, blue) is rotated by multiples of $\theta$ ($4 \times \theta$) to produce the rotated vector (RV, red), illustrating how cyclic SO(2) subgroup embeddings differentiate synonymous codons while preserving amino acid equivariance.

Our contributions are summarized as follows:

- We introduce **Equi-mRNA**, the first mRNA language model to explicitly encode codon-level symmetries via group-theoretic priors, enforcing biologically grounded SO(2) equivariance with an auxiliary loss and symmetry-aware pooling mechanisms.

- We curate and release a unified coding-region corpus of **25M** protein-coding sequences plus a stratified **1M** sequence subset to standardize benchmarking across studies.

- Equi-mRNA achieves **up to $\approx$ 10% improvements in downstream property prediction**, and **up to $\approx$ 4× more realistic mRNA sequence generation** with $\approx$ **28% better preservation of functional properties** compared to vanilla baselines.
- Through interpretability analyses, we link learned codon-rotation distributions to GC-content biases and tRNA abundances, and demonstrate that Equi-mRNA outperforms state-of-the-art nucleotide and codon models with fewer parameters.

## 2 Methodology

In this section, we will introduce the group-theoretic embedding framework for codon-level language models. We will first describe the mathematical structure of the genetic code and how it can be represented as a group. Then, we will outline the embedding process and how it can be integrated into existing language model architectures.

### 2.1 Problem Formulation

Let $\mathcal{C}$ denote the set of 64 codons (including start and stop), and let $\mathcal{A}$ be the set of 20 amino acids together with a stop symbol. The genetic code defines a surjective map $\pi : \mathcal{C} \to \mathcal{A}$, where codons mapping to the same amino acid are considered synonymous. For each $a \in \mathcal{A}$, define $\mathcal{C}_a = \pi^{-1}(a)$ as the corresponding synonym set, and let $n_a = |\mathcal{C}_a|$. By construction, the sets $\mathcal{C}_a : a \in \mathcal{A}$ form a disjoint partition of $\mathcal{C}$. For preliminaries and notations please refer to Appendix A.1.

**Cyclic action on synonymous codons.** Fix an amino acid $a$ with its set of synonymous codons $\mathcal{C}_a = \{c_{a,0}, \dots, c_{a,n_a-1}\}$ in a chosen order. Let $G_a = \mathbb{Z}_{n_a}$ be the finite cyclic group with addition modulo $n_a$. Define a group action $\varphi_a : G_a \times \mathcal{C}_a \to \mathcal{C}_a$ by

$$\varphi_a(k,\, c_{a,i}) \;=\; c_{a,\,(i+k) \bmod n_a}. \tag{1}$$

Here the group is $G_a$, a group element is $k \in G_a$, and a generator is any $g_a \in G_a$ such that repeated application of $\varphi_a(g_a, \cdot)$ visits every codon in $\mathcal{C}_a$. This action is regular, it is free and transitive on $\mathcal{C}_a$. When $n_a = 1$, the action is trivial. This encoding treats each synonym set as a discrete symmetry class that reflects the degeneracy of the genetic code [22].

**Definition 2.1.** *(Synonymous Codon Group). For each amino acid $a \in \mathcal{A}$ with $n_a = |\mathcal{C}_a| > 1$, let $\varphi_a : \mathcal{C}_a \to 0, 1, \dots, n_a - 1$ be an arbitrary bijection that assigns a unique integer label to each codon in $\mathcal{C}_a$. We induce a cyclic group structure on $\mathcal{C}_a$ by identifying it with $\mathbb{Z}_{n_a}$ via $\varphi_a$, where the group operation is defined as addition modulo $n_a$.*

Under Definition 2.1, index the codons by a labeling map $\ell_a : \mathcal{C}_a \to \mathbb{Z}_{n_a}$ with $\ell_a(c_{a,i}) = i$. The identity element corresponds to the codon with label 0. Let $1 \in G_a = \mathbb{Z}_{n_a}$ be a generator. We define the *canonical substitution step* as the action of 1 on any codon, that is,

$$c_{a,i} \;\mapsto\; \varphi_a(1,\, c_{a,i}) \;=\; c_{a,\,(i+1) \bmod n_a},$$

which replaces a codon by the next synonymous codon in the fixed cyclic order. Repeated application of this step enumerates all codons in $\mathcal{C}_a$, and the inverse step uses the action of $-1$ to move to the previous codon.

This construction yields a well-defined group action of $G_a \cong \mathbb{Z}_{n_a}$ on $\mathcal{C}_a$, where substituting codon $c \in \mathcal{C}_a$ with another synonymous codon $c'$ corresponds to applying an element $j \in \mathbb{Z}_{n_a}$: if $\varphi_a(c) = k$, then $\varphi_a(c') = k + j \mod n_a$. Thus, synonymous codon substitutions are formalized as actions of a finite cyclic group. Our objective is to build codon embeddings that explicitly respect this group structure.

### 2.2 Mapping Codons to SO(2) via Cyclic Subgroups

While the codon group structure (Definition 2.1) captures the finite cyclic symmetry within each synonym set, it is non differentiable and thus incompatible with gradient based learning. To address this, we define a continuous group homomorphism $\rho_a : G_a \to \mathrm{SO}(2)$ that maps $k \in \mathbb{Z}_{n_a}$ to a planar rotation with $\theta_a = 2\pi/n_a$ or a learned angle constrained to preserve the homomorphism. This preserves the cyclic structure while embedding codons in a differentiable manifold suitable for neural models. Although the genetic code does not inherently suggest a rotational structure, the use of $\mathrm{SO}(2)$ serves as an inductive bias that reflects codon-level redundancy through structured, parameter-sharing representations. Its compact topology and continuous symmetry enable smooth optimization, enforce equivariant constraints, and promote generalization across contexts.

**Construction 2.2.** *Let $\theta_a := \frac{2\pi}{n_a}$ denote the angular increment corresponding to one step in the cyclic group $G_a$. We define a homomorphism $\Phi$ from codons to $SO(2)$ by mapping each codon $c \in \mathcal{C}_a$ to a rotation matrix:*

$$\Phi(c) := R\left(\varphi_a(c) \cdot \theta_a\right) = \begin{pmatrix} \cos(\varphi_a(c)\theta_a) & -\sin(\varphi_a(c)\theta_a) \\ \sin(\varphi_a(c)\theta_a) & \cos(\varphi_a(c)\theta_a) \end{pmatrix}.$$

*The image of this mapping is the set of $n_a$ evenly spaced rotations:*

$$\{R(0), R(\theta_a), R(2\theta_a), \ldots, R((n_a - 1)\theta_a)\},$$

*which forms a cyclic subgroup of $SO(2)$ isomorphic to $\mathbb{Z}_{n_a}$. The generator $g_a = \varphi_a^{-1}(1)$ is mapped to $R(\theta_a)$, and we have $\Phi(g_a^{n_a}) = R(n_a \cdot \theta_a) = R(2\pi) = I$, consistent with the identity element in both groups.*

As formalized in Proposition A.1 (Appendix A.6), the map $\Phi_{\theta_a}$ defines a group homomorphism from the codon substitution group $G_a \cong \mathbb{Z}_{n_a}$ to the cyclic subgroup of $SO(2)$ generated by $R(\theta_a)$. This ensures that synonymous codon substitutions correspond to structured, differentiable rotations in the embedding space.

**Example** Serine is encoded by $n_{\text{Ser}} = 6$ codons:

$$\mathcal{C}_{\text{Ser}} = \{\text{UCU}, \text{UCC}, \text{UCA}, \text{UCG}, \text{AGU}, \text{AGC}\},$$

which we label via $\varphi_{\text{Ser}} = \{0, 1, 2, 3, 4, 5\}$. Under $\Phi$, these codons are mapped to:

$$\{R(0°), R(60°), R(120°), R(180°), R(240°), R(300°)\},$$

i.e., the vertices of a regular hexagon on the unit circle. Substituting UCU (label 0) by AGC (label 5) corresponds to applying the group action five times: $\Phi(\text{AGC}) = R(5\theta_{\text{Ser}})\,\Phi(\text{UCU}) = R(-\theta_{\text{Ser}})\,\Phi(\text{UCU})$ modulo $2\pi$. Thus, $\mathcal{C}_{\text{Ser}}$ is embedded in a cyclic subgroup of $SO(2)$ of order 6. Since $SO(2)$ is a continuous Lie group, this embedding gives the discrete $\mathbb{Z}_6$ symmetry a smooth, differentiable representation, making it suitable for integration with gradient based learning.

## 2.3 Learning Codon Rotation Generators: Toward Adaptive and Robust Representations

The canonical construction maps each synonym set $\mathcal{C}_a$ to a cyclic subgroup of $SO(2)$ by fixing a generator angle $\theta_a = \frac{2\pi}{n_a}$. This produces uniform rotations that exactly realize the $\mathbb{Z}_{n_a}$ action, but it also fixes the geometry a priori and may fail to capture organism specific or condition specific variation.

To retain differentiability while allowing data driven adaptation, we use learnable parameterizations that preserve the group structure either exactly or approximately. A strict homomorphic parameterization sets and learns $\theta_a$ under the constraint $\rho_a(n_a) = I$. This is enforced either by parameterizing $\theta_a = \frac{2\pi m_a}{n_a}$ with $m_a$ an integer coprime to $n_a$, or by adding a penalty $\lambda \left\| \rho_a(n_a) - I \right\|_F^2$ during training. For higher dimensional embeddings, we lift the action to learned 2D subspaces by introducing an orthonormal basis $U_a \in \mathbb{R}^{d \times 2}$ and applying

$$x \;\mapsto\; x + U_a\big(R(k\,\theta_a) - I_2\big)U_a^\top x,$$

which rotates only the codon specific plane while leaving the orthogonal complement unchanged. These parameterizations keep the cyclic symmetry visible to the model, remain fully differentiable, and give the geometry enough freedom to reflect biological variability across datasets.

To overcome this limitation, we propose to treat the generator angle $\theta_a \in \mathbb{R}$ as a learnable parameter for each amino acid group. The codon embedding map is then defined as

$$\Phi_{\theta_a}(c) = R\left(\varphi_a(c) \cdot \theta_a\right),$$

where $\theta_a$ is optimized jointly with the model. Crucially, the generality of Proposition A.1 ensures that the group homomorphism property continues to hold under this parameterization, provided that $\theta_a \cdot n_a \equiv 0 \mod 2\pi$, which can be softly enforced or reparameterized during training.

Learning $\theta_a$ introduces only one additional parameter per amino acid and provides a biologically grounded, symmetry-preserving mechanism for fine-tuning codon embeddings. Unlike traditional fixed-vector approaches, it maintains consistent relative geometry among synonymous codons while enabling global adaptation to species-specific or context-dependent signals. This structured update mechanism improves flexibility and stability during downstream adaptation, offering a principled and efficient alternative to conventional embeddings.

**Higher-dimensional extensions.** To increase representational capacity, we also extend this construction to higher-dimensional embeddings via block-diagonal rotations in $SO(d)$, enabling multiple independent subspaces per codon. This generalization preserves group-theoretic structure while allowing richer task-specific codon representations. Full details of this extension and its formal properties are presented in Appendix A.3.

## 2.4 Fuzzy Codon Embeddings: Soft Group Actions from Learned Distributions

Strict cyclic embeddings assume that each codon corresponds to a fixed rotation derived from a group generator. While this enforces clean algebraic structure, it may be too rigid in biological settings where codon preferences are not binary but graded, noisy, or context-dependent. To address this, we introduce a fuzzy relaxation of the codon-to-rotation mapping that allows each codon to induce a distribution over rotation angles, rather than a fixed one.

In our fuzzy formulation, each amino acid group has a learnable angle distribution over a fixed number $K$ of discrete rotation prototypes. These prototypes can be uniform (e.g., evenly spaced over $[0, 2\pi)$) or trainable. For each codon, we compute a softmax over the angle logits `angle_logits` $\in \mathbb{R}^{|\mathcal{A}| \times m \times K}$, yielding a smooth distribution over $K$ angular components. The effective codon angle is then defined as a weighted average over angle bins:

$$\theta(c) = \frac{2\pi}{k_a} \sum_{j=0}^{K-1} p_{c,j} \cdot j,$$

where $p_{c,j}$ is the softmax probability assigned to codon $c$ for angle bin $j$, and $k_a$ is the number of synonymous codons for amino acid $a$. This angle is then scaled by the codon's group label to form the final rotation: $\varphi_a(c) \cdot \theta(c)$.

This fuzzy formulation retains the inductive bias of cyclic symmetry while allowing biologically meaningful deviations, such as soft substitutions or asymmetric codon effects. It supports gradient-based optimization over both rotation parameters and angle distributions, making it amenable to fine-tuning and transfer. In the limit, it recovers the hard cyclic embedding, allowing the model to interpolate between strict and relaxed symmetry in a task-adaptive manner.

## 2.5 General Basis Rotation via Stiefel-Manifold-Constrained Subspaces

We extend codon embeddings by representing each synonymous codon as a rotation of a shared base embedding vector within a learned 2D subspace of $\mathbb{R}^d$. We work in an ambient space $\mathbb{R}^d$. For each amino acid $a$, let $U_a \in \mathbb{R}^{d \times 2}$ have orthonormal columns that span a learned two dimensional subspace, and let $e_a \in \mathbb{R}^d$ be a shared base embedding. A synonymous codon with angle $\theta_{a,k}$ is represented by

$$x_{a,k} = \left(I - U_a U_a^\top\right) e_a + U_a R(\theta_{a,k}) U_a^\top e_a, \qquad R(\theta) = \begin{bmatrix} \cos\theta & -\sin\theta \\ \sin\theta & \cos\theta \end{bmatrix}.$$

Thus the embedding lives in $\mathbb{R}^d$, the rotation acts only in the learned two dimensional subspace, and the orthogonal complement is preserved.

For each amino acid $A$, let $\mathbf{z}_A \in \mathbb{R}^d$ denote a shared base embedding vector. We define a rotation subspace for $A$ spanned by an orthonormal pair of vectors $(\mathbf{u}_A, \mathbf{v}_A) \in \mathbb{R}^d$, where $\mathbf{u}_A := \mathbf{z}_A / \|\mathbf{z}_A\|$ is aligned with the direction of the base vector. The second vector $\mathbf{v}_A \in \mathbb{R}^d$ is constrained such that $\langle \mathbf{u}_A, \mathbf{v}_A \rangle = 0$ and $\|\mathbf{v}_A\| = 1$, forming an orthonormal frame in $\mathbb{R}^d$. We model this 2D subspace using the *Stiefel manifold* $\mathrm{St}(2, d)$, which is the space of all orthonormal 2-frames in $\mathbb{R}^d$.

Each codon $c \in \mathcal{C}_A$ is associated with a rotation angle $\phi_c \in [0, 2\pi)$, and its embedding is computed via:
$$\mathbf{E}(c) = R_{(\mathbf{u}_A, \mathbf{v}_A)}(\phi_c)\, \mathbf{z}_A = \cos\phi_c\, \mathbf{z}_A + \sin\phi_c\, \|\mathbf{z}_A\| \mathbf{v}_A,$$

where $R_{(\mathbf{u}_A, \mathbf{v}_A)}(\phi_c)$ denotes the rotation in the plane spanned by $\{\mathbf{u}_A, \mathbf{v}_A\}$ and acts as identity on the orthogonal complement. This embedding guarantees that all synonymous codons of $A$ lie on a circle of radius $\|\mathbf{z}_A\|$ within a task-learned semantic plane, enabling flexible and structured encoding of codon variation.

To ensure the basis $(\mathbf{u}_A, \mathbf{v}_A)$ remains orthonormal during training, we parameterize the subspace using a point on the Stiefel manifold and optimize it via Riemannian gradient descent. In practice,

we implement this using the geoopt library, which constrains each learned matrix $V_A \in \mathbb{R}^{d \times 2}$ such that $V_A^\top V_A = I$. The rotated codon embeddings can then be expressed as:

$$\mathbf{E}(c) = V_A \cdot R(\phi_c) \cdot V_A^\top \mathbf{z}_A,$$

where $R(\phi_c) \in SO(2)$ is the canonical planar rotation matrix and $V_A \in \mathrm{St}(2, d)$. This formulation naturally supports group-equivariant representations while permitting the geometry of codon substitution to be learned from data (see Proposition A.3).

### 2.6 Equivariance Enforcement via Auxiliary Loss

To ensure that codon-level symmetries are preserved beyond the embedding layer, we incorporate an *auxiliary equivariance loss* that encourages internal representations to transform consistently under synonymous codon substitutions. This regularization promotes structured and predictable behavior aligned with the underlying group action, improving robustness, generalization to unseen codon contexts, and interpretability of the learned representations. It also facilitates stable fine-tuning of rotation generators $\theta_a$ in transfer settings, ensuring that codon embedding geometry adapts smoothly to species- or task-specific preferences. The full formulation, training objective, and mathematical properties of this loss are provided in Appendix A.4.

### 2.7 Equivariant Pooling Mechanisms for SO(2) Representations

To maintain architectural consistency with the $SO(2)$-equivariant codon embeddings, we implement specialized pooling mechanisms that preserve group structure during sequence-level aggregation. These include polar pooling, Fourier-based pooling, and direct angular averaging, each designed to respect the rotational symmetries inherent in the embedding space. For a full mathematical formulation and biological motivation of these $SO(2)$-aware pooling strategies, please refer to Appendix A.5.

## 3 Experiments

### 3.1 Experimental Setup

**Pretraining Corpus**  We constructed a large-scale pretraining corpus by drawing 25 million annotated protein-coding sequences from 56 million RefSeq entries and retaining only those between 20 and 512 codons in length. Untranslated flanking regions were discarded to focus on genuine coding signals. From this filtered set we sampled a stratified 1 million-sequence subset (preserving original taxonomic proportions) for controlled ablations, then used the full 25 million-sequence collection to train our final Equi-mRNA variant at scale (Appendix A.9.1).

**Downstream Tasks**  We evaluate on six biologically driven benchmarks spanning expression, stability, and regulatory switching (MLOS [23], mRFP [29], E. coli expression [10], Tc-riboswitch switching factor [15], iCodon stability [9], SARS-CoV-2 degradation [48]; see Appendix A.9.2 for details).

**Two-Stage Ablation Protocol**  We begin with an extensive ablation on the 1 M-sequence subset randomly sampled from the curated corpus to assess the impact of three embedding parameterizations: Fixed, Learned, and Fuzzy $\theta$. Each is combined with or without the Stiefel rotation basis and with or without the equivariance loss $\mathcal{L}_{\mathrm{equiv}}$, yielding 12 total variants. Table 3 summarizes these models, and Figure 4 (Appendix A.2) provides a schematic overview. All variants share an identical GPT2 Transformer backbone [37], ensuring that performance differences arise solely from embedding design. Note that the Stiefel and Fuzzy variants introduce additional parameters to capture their geometric transforms.

**Full-Scale Training**  Following ablation, we selected the top-performing configuration and retrained it on the full 25 M-sequence corpus to evaluate scalability and generalization under abundant data. Pretraining was conducted on thirty-two NVIDIA H100 GPUs (ablation used eight NVIDIA H200 GPUs); runtimes and resource utilization are detailed in Appendix A.11.

**Hyperparameters**  All pretraining arguments and hyperparameters, as well as downstream generation and property-prediction hyperparameters, are provided in Appendix A.10.

### 3.2 Symmetry-Aware Property Prediction

We summarize the downstream performance of our twelve symmetry-aware variants across five biological benchmarks. Figure 1a shows the average accuracy and Spearman correlation improvements

Table 1: Component-wise comparison of symmetry-aware variants: Equivariant SO(2) models improve downstream accuracy over vanilla across all tasks. Fuzzy and Learned $\theta$ variants with Stiefel basis and/or equivariant loss yield the best results, highlighting the benefit of structured group-theoretic priors.

| Model | Stiefel | Equiv. | E. coli(A) | MLOS(S) | Tc-Ribo.(S) | mRFP(S) | COV Deg(S) |
|-------|---------|--------|------------|---------|-------------|---------|------------|
| Vanilla | - | - | 0.580 | $0.633 \pm 0.14$ | 0.698 | 0.797 | 0.779 |
| Fixed $\theta$ | ✓ | ✗ | 0.602 | $0.667 \pm 0.19$ | 0.688 | 0.840 | 0.803 |
| Learned $\theta$ | ✓ | ✗ | **0.633** | $0.657 \pm 0.10$ | 0.701 | **0.871** | 0.790 |
| Fuzzy $\theta$ | ✓ | ✓ | *0.605* | **0.691 $\pm$ 0.14** | **0.736** | *0.844* | **0.820** |

over the vanilla baseline for each rotation strategy and symmetry constraint. To highlight overall trends, Table 1 reports the top-performing configuration from each embedding group (fixed, learned, and fuzzy $\theta$). The fuzzy $\theta$ model with Stiefel basis and equivariant loss achieves the highest gains overall and was therefore selected for large-scale training on the full 25M sequence corpus. From this point forward, we refer to this fuzzy $\theta$ Stiefel with equivariance configuration trained on the 25M sequence corpus as EQUI-MRNA Model. Full results for all variants and evaluation metrics are provided in Appendix A.12. Note that the MLOS dataset lacks standard splits and exhibits high variability due to its small size; we report standard deviations accordingly to reflect this sensitivity.

An in-depth component analysis in Table 14 (Appendix A.12) isolates the contribution of each design element. Fixed-angle models with equivariance already outperform the vanilla baseline on *E. coli* and *mRFP* (0.600 and 0.841 vs. 0.580 and 0.797), demonstrating that even static geometric priors can enhance biological alignment. Incorporating learned $\theta$ with a Stiefel projection further boosts performance, yielding the highest accuracy on *E. coli* (0.633) and peak Spearman correlation on *mRFP* (0.871), which suggests that data-driven codon embeddings capture species or tissue-specific regulatory signals. Equivariant training adds additional gains on structurally sensitive tasks such as *Tc-Riboswitch* (0.741) and *MLOS* long sequences (0.698). Finally, fuzzy $\theta$ models representing codons as soft distributions over SO(2) deliver robust improvements on noisy or low-resource assays like *MLOS* (0.691) and *SARS-CoV-2 degradation* (0.820), further motivating their adoption for full-scale training.

Table 2 compares our EQUI-MRNA models (5M and 15M parameters) to established codon and nucleotide-based baselines trained on the full 25M-sequence corpus. The 15M EQUI-MRNA variant achieves the highest accuracy across all evaluated tasks while using only $\approx 30\%$ of the parameters of HELM. Even the smaller 5M model matches or surpasses several larger architectures, highlighting the efficiency of our symmetry-aware approach. While exact comparisons are limited by differences in pretraining protocols, objectives, and the public availability of models (e.g., HELM's pre-training data is not publicly released), these results underscore the effectiveness of embedding codon symmetries directly. Specifically, our 5M parameter variant of EQUI-MRNA employs a hybrid Mamba–Transformer backbone, demonstrating that symmetry-aware SO(2) embeddings can achieve competitive performance with minimal overhead and be seamlessly extended to architectures such as state-space models.

### 3.3 Symmetry-Aware Sequence Generation

We conducted our experiments following the framework established by [52]. For generative evaluation, we used the iCodon thermostability corpus: up to 1 000 test sequences were selected and truncated after two-thirds of their length, with the prefix serving as a prompt for autoregressive completion of the remaining one-third via codon sampling ($k = 10$, $p = 0.95$) at temperatures $T \in \{0.2, 0.4, 0.6, 0.8, 1.0\}$. Generative quality was quantified by Fréchet BioDistance (FBD) between synthetic and true suffixes, computed as in [46]. All FBD scores for equivariant variants appear in Figure 2, with complete results in Appendix A.13.1.

To evaluate retention of functional properties, we assessed generated sequences on five downstream benchmarks (iCodon, MLOS, mRFP, Tc-Riboswitch, and SARS-CoV-2 degradation). For each dataset, up to 1 000 sequences were truncated after two-third of their length, and the remaining one-thirds were generated under the same codon sampling protocol. We then applied pre-trained CodonBert-based property prediction model [23] to estimate task-specific labels and computed mean squared error (MSE) between predicted and true labels; lower MSE indicates better preservation of

Table 2: Performance comparison across downstream benchmarks: Equi-mRNA models (trained on 25M sequences) outperform prior nucleotide and codon-based baselines across six biological tasks. The 15M GPT-2 variant achieves the highest accuracy in 5 out of 6 datasets, demonstrating the effectiveness of symmetry-aware embeddings at scale.

| Model | E. coli(A) | MLOS(S) | iCodon(S) | Tc-Ribo.(S) | mRFP(S) | COV Deg(S) |
|---|---|---|---|---|---|---|
| *Nucleotide-Based* | | | | | | |
| RNA-FM | 0.43 | - | 0.34 | 0.58 | 0.80 | 0.74 |
| RNABERT (82 M) | 0.39 | - | 0.16 | 0.47 | 0.40 | 0.64 |
| Aido mRNA (1.6B) | 0.576 | $0.504 \pm 0.23$ | 0.472 | 0.492 | 0.683 | 0.743 |
| CALM | - | $0.430 \pm 0.170$ | 0.376 | 0.625 | 0.546 | 0.773 |
| mRNA-FM | - | $0.509 \pm 0.154$ | 0.458 | 0.690 | 0.564 | 0.714 |
| *Codon-Based* | | | | | | |
| CodonBert (82 M) | 0.57 | 0.543 | 0.350 | 0.502 | 0.832 | 0.78 |
| GPT2 (CLM)(50M)[*] | - | 0.611 | 0.498 | 0.531 | 0.815 | 0.787 |
| GPT2 (MLM)(50M)[*] | - | 0.653 | 0.503 | 0.569 | 0.753 | *0.801* |
| HELM (CLM)(50M)[*] | - | 0.592 | 0.529 | 0.619 | 0.849 | 0.789 |
| HELM (MLM)(50M)[*] | - | 0.701 | 0.525 | 0.626 | 0.822 | **0.833** |
| **Equi-mRNA (5M)**[†] | *0.581* | *0.705 ± 0.12* | 0.519 | **0.764** | *0.853* | 0.756 |
| **Equi-mRNA (15M)**[‡] | **0.613** | **0.710 ± 0.13** | **0.537** | *0.737* | **0.855** | 0.791 |

[†] Mamba-based hybrid Architecture; [‡] GPT-2 Architecture both trained on 25M datapoints

[*] Trained on Antibody mRNA sequences

biological function. All MSE reduction compared to Vanilla model for equivariant variants appear in Figure 3, with complete results in Appendix A.13.2.

### 3.3.1 Enhanced Generative Fidelity with Equi-mRNA

Analyzing the generated sequences quality results, we observe a stark contrast between the vanilla GPT-2 baseline and our symmetry-aware models. The baseline's FBD increases with temperature indicating instability; whereas all equivariant variants exhibit a smooth, pronounced decline in FBD as $T$ rises. The best performer the equivariant Fuzzy $\theta$ model achieves an FBD of $\approx 580$ at $T = 1.0$, a roughly **4.3 times** improvement over vanilla. Equivariant Fixed $\theta$ and Learned $\theta$ converge near $10^3$, about a **threefold gain**.

The sharp but smooth decline in FBD indicates that equivariant models increasingly capture latent sequence modes absent in the baseline but prevalent in real data. Rather than degenerating into noise, higher temperatures amplify those syntactic alternatives

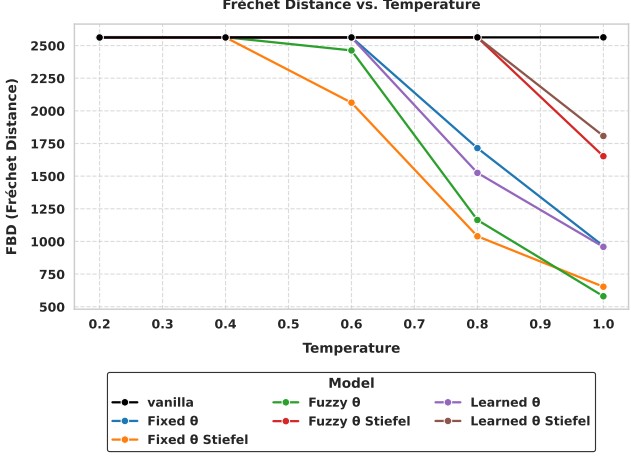

Figure 2: FBD vs. temperature for equivariant SO(2) variants and Vanilla model: FBD between generated and true suffixes on iCodon improves (lower ↓) with temperature for all SO(2)-equivariant variants, unlike the vanilla model which degrades. Fuzzy $\theta$ achieves the best score (∼580 at $T$=1.0), a 4.3× gain over baseline.

permitted by the group-theoretic prior, reducing distributional shift without sacrificing biological plausibility. In this way, symmetry-aware architectures and equivariance regularization convert temperature from a benign rescaling constant into a principled mechanism for structured sequence exploration.

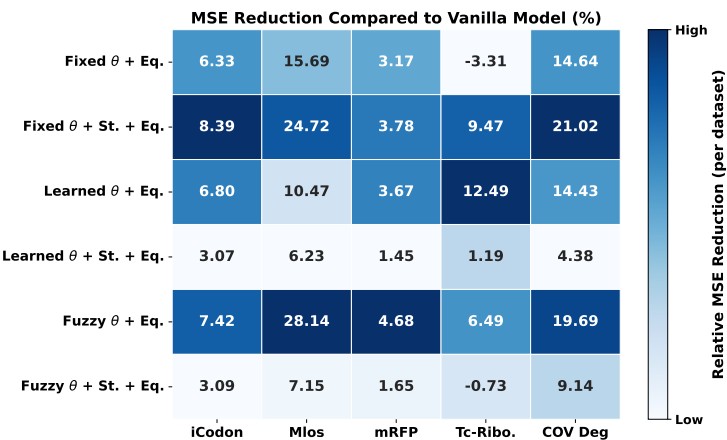

Figure 3: MSE reduction for equivariant SO(2) variants vs. vanilla: SO(2)-equivariant variants consistently reduce MSE (lower ↓) in property prediction across all datasets, with up to 28% improvement over Vanilla model. Results reflect enhanced biological plausibility and downstream utility.

Lastly, scaling pretraining from 1M to 25M sequences without any architectural modifications drives FBD down to **76.13** for the Equi-mRNA (5M) variant and **177.77** for Equi-mRNA (15M), representing a significant leap forward in generative fidelity at this scale. These results not only demonstrate the remarkable scalability and robustness of symmetry-aware embeddings but also establish a new state of the art for large-scale biological sequence generation.

### 3.3.2 Enhanced Property Retention under Equi-mRNA Embeddings

Beyond sequence quality, we assessed property prediction by training a CodonBert-based regression model on each dataset and measuring MSE [23]. Figure 3 demonstrates that equivariant SO(2) models consistently lower MSE relative to vanilla GPT-2 across all benchmarks achieving up to a 28% reduction in error. Notably, the MLOS dataset, though small (<1000 sequences), comprises $\approx$ 1000 codon mRNAs, while Tc-Riboswitch sequences span only 26 codons; both contexts yield uniform MSE improvements under equivariance. Such a substantial decrease in prediction error not only confirms the capacity of structured SO(2) embeddings to capture rotational symmetries and long-range dependencies, but also translates directly into more faithful preservation of functional properties. In practice, this level of accuracy can significantly reduce the cost and time of experimental validation by prioritizing higher-quality candidate sequences. For brevity, only equivariant-enforced variant results are shown here; full comparisons among fixed, learned, and fuzzy $\theta$ models are provided in Appendix A.13.2.

### 3.4 Interpretability Analysis

To test whether the imposed $SO(2)$ structure yields biologically meaningful representations *within a single species*, we fine tuned a 5M parameter model on the human coding transcriptome (GRCh38) [27] and reserved 20% for evaluation. Focusing on human controls for cross species confounders, lets us probe a consistent codon usage bias, and provides a clear path to codon optimization policies that are specific to one organism rather than averaged across taxa. The goal is not only to interpret embeddings after the fact, but to validate that codon level rotational geometry captures constraints that matter for human design and optimization.

First, we quantified the uncertainty of learned codon angles using Shannon entropy and analyzed its dependence on transcript GC content, which is a known correlate of secondary structure and expression control. Binning sequences by GC proportion revealed a near linear trend in mean angle entropy ($r = 0.98$, $R^2 = 0.97$, $p < 10^{-11}$), suggesting that GC rich regions induce more uncertain codon rotations. This behavior is consistent with our parameterization, where the distribution over angles can encode biological variability.

Second, we examined whether the codon embeddings capture translational supply constraints by correlating learned angles with normalized human tRNA gene counts [18]. We selected the subspace block most predictive of tRNA levels and computed a weighted composite angle per codon. A Spearman correlation of $\rho = -0.69$ indicates that codons with higher tRNA availability are systematically

assigned smaller angles, consistent with efficient translation demands being reflected in the rotational geometry.

These results support the view that the group theoretic prior is both mathematically sound and aligned with human specific biology. Visualizations and further details are provided in Appendix A.8.

# 4    Limitation and Future works

While Equi-mRNA introduces biologically grounded inductive biases by enforcing cyclic SO(2) rotations over synonymous codons, it is currently constrained to protein coding regions and fixed triplet tokenization. This design overlooks non-coding elements such as untranslated regions (UTRs) and may obscure gene or species-specific codon usage patterns. Furthermore, the incorporation of fuzzy angle distributions and Stiefel manifold rotations enhance representational flexibility but incurs additional parameterization and Riemannian optimization overhead, potentially hindering scalability in resource-constrained or large-scale settings. Current evaluations are limited to a narrow set of benchmarks, leaving the model's robustness on long transcripts, non-standard GC content, non-model organisms, and real-world sequence design tasks largely unexplored.

Future extensions may benefit from meta-learning codon-specific priors using hyper-networks or generative modules that adapt rotation parameters dynamically across organisms, tissues, or experimental regimes with minimal fine-tuning. Additionally, exploring richer group-theoretic structures such as non-abelian or product groups could enable modeling of local codon interactions, frame-shift motifs, or positional dependencies. Expanding empirical validation to more diverse settings and practical applications will be essential for assessing generalization and informing deployment under realistic biological constraints.

# 5    Conclusion

We introduced Equi-mRNA, a novel mRNA language modeling framework that embeds codon-level symmetries as differentiable group actions within the SO(2) Lie group. By aligning the genetic code's redundancy with structured geometric priors, our model captures synonymous codon relationships through rotation-equivariant embeddings, auxiliary symmetry-enforcing losses, and biologically motivated pooling mechanisms. Across diverse benchmarks, Equi-mRNA consistently improves downstream property prediction accuracy and generative fidelity, achieving up to ~10% accuracy gains and $4.3\times$ reductions in Fréchet BioDistance over strong baselines.

Beyond empirical performance, our interpretability analysis reveals that learned codon rotation patterns align with known GC-content and tRNA abundance biases, offering biologically grounded insights into translation regulation. Together, these results establish Equi-mRNA as a scalable, interpretable, and biologically faithful foundation for modeling protein-coding sequences, with broad implications for synthetic mRNA design and genomic modeling. Future work will extend this symmetry-aware paradigm to non-coding regions, richer group structures, and adaptive priors for cross-species generalization.

## Acknowledgments

This work used DeltaAI at NCSA through allocation CIS250398 from the Advanced Cyberinfrastructure Coordination Ecosystem: Services & Support (ACCESS) program, which is supported by U.S. National Science Foundation grants #2138259, #2138286, #2138307, #2137603, and #2138296 [4].

We thank Dr. Ivan Garibay for his thorough feedback on the manuscript and ongoing support throughout this research.

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

# A   Technical Appendices and Supplementary Material

## A.1   Preliminaries

This section introduces the algebraic and geometric foundations that motivate our model design, particularly the use of cyclic groups, the special orthogonal group $\mathrm{SO}(2)$, and the Stiefel manifold. These constructs enable biologically grounded and differentiable representations of codon symmetries in mRNA sequences.

**Group and Cyclic Groups.**   A *group* is a set $G$ equipped with a binary operation $\cdot$ satisfying closure, associativity, identity, and invertibility. A group is called *cyclic* if there exists a generator $g \in G$ such that every element in $G$ can be written as $g^k$ for some integer $k$. The finite cyclic group of order $n$ is denoted $\mathbb{Z}_n$, with addition modulo $n$ as its operation.

**Group Action.**   A *group action* of $G$ on a set $X$ is a function $G \times X \to X$ such that the identity acts as the identity transformation on $X$, and the action respects the group composition. In our context, synonymous codons form equivalence classes $C_a \subset C$, and the group $\mathbb{Z}_{n_a}$ acts on $C_a$ by cyclically permuting codons.

**Lie Groups and $\mathrm{SO}(2)$.**   A *Lie group* is a smooth manifold endowed with a group structure. The group $\mathrm{SO}(2) \subset \mathbb{R}^{2\times 2}$ consists of 2D rotation matrices of the form

$$R(\theta) = \begin{pmatrix} \cos\theta & -\sin\theta \\ \sin\theta & \cos\theta \end{pmatrix},$$

forming a compact and differentiable group under matrix multiplication. We construct a homomorphism $\Phi : \mathbb{Z}_{n_a} \to \mathrm{SO}(2)$ by mapping codon labels to evenly spaced angles $\theta_k = 2\pi k / n_a$, enabling gradient-based learning on codon symmetries.

**Stiefel Manifolds.**   The *Stiefel manifold* $\mathrm{St}(k, d)$ is the set of all orthonormal $k$-frames in $\mathbb{R}^d$, i.e.,

$$\mathrm{St}(k, d) = \left\{ V \in \mathbb{R}^{d\times k} \mid V^\top V = I_k \right\}.$$

This manifold defines a smooth subspace of $\mathbb{R}^{d\times k}$, with non-Euclidean geometry, and is central to our parameterization of codon rotation subspaces. In Equi-mRNA, each amino acid defines a local 2D subspace spanned by an orthonormal frame $(u_A, v_A) \in \mathrm{St}(2, d)$, within which synonymous codons are rotated.

## A.2   Overview of Codon Embedding Variants

To complement the detailed description in the main text, Figure 4 provides a unified visual summary of our codon embedding framework and its key variants. Each model encodes synonymous codons as rotated versions of a shared amino acid base embedding, differing along three key design axes: the rotation generator (fixed, learned/cyclic, or fuzzy), and the rotation basis (standard 2D plane or learned Stiefel subspace). This schematic illustrates the modular structure of the embedding pipeline and how group-theoretic principles govern the construction of codon-level representations across all twelve variants evaluated in our experiments.

## A.3   Block-Diagonal Codon Embeddings in Higher Dimensions

While $SO(2)$ suffices to model codon substitution as a planar rotation, it is often beneficial in practice to embed codons in higher-dimensional vector spaces. To preserve the group-theoretic structure while scaling up the representation space, we extend the $SO(2)$ embedding to a block-diagonal $SO(d)$ representation, where $d$ is even.

We partition each codon embedding vector into $d/2$ disjoint 2D subspaces. Each subspace is then rotated independently by an angle $\theta_a^{(i)} \in \mathbb{R}$, where $i = 1, \ldots, d/2$ indexes the sub-blocks, and $\theta_a^{(i)}$ is a learnable generator parameter specific to amino acid $a$. Let $\Theta_a = (\theta_a^{(1)}, \ldots, \theta_a^{(d/2)}) \in \mathbb{R}^{d/2}$ denote the full set of learned rotation generators for amino acid $a$.

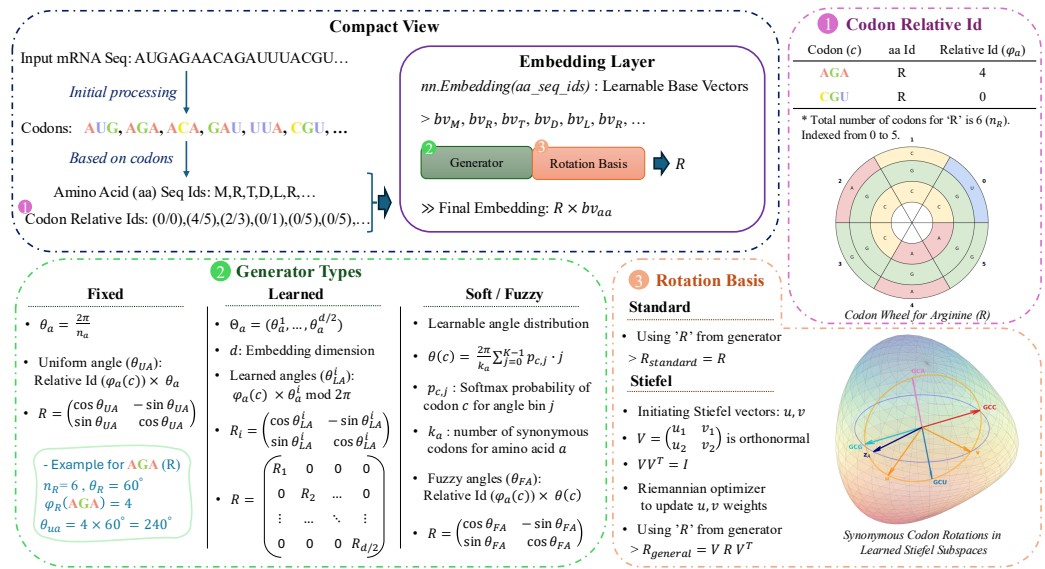

Figure 4: Illustration of our codon embedding framework and the key variants evaluated in this study. Each codon is mapped to a rotated copy of its amino acid base vector, using either fixed, learned (cyclic), or fuzzy angle generators. Rotation is applied in either a fixed 2D plane or a learned subspace on the Stiefel manifold. This figure summarizes the geometric mechanisms underlying the twelve variants listed in Table 3.

Table 3: Overview of codon embedding variants studied. Each can be trained with or without the equivariance loss. "Standard" uses a fixed 2D subspace, "General" uses a learned Stiefel manifold basis. "Cyclic" uses learned generator angles, "Fuzzy" uses soft angle distributions.

| Model Name | Rotation Basis | Generator Type |
|---|---|---|
| SO2-fixed-standard | Standard (fixed) | Fixed generator (uniform angles) |
| SO2-fixed-general | Learned basis (Stiefel) | Fixed generator (uniform angles) |
| SO2-cyclic-standard | Standard (fixed) | Learned generator (cyclic angles) |
| SO2-cyclic-general | Learned basis (Stiefel) | Learned generator (cyclic angles) |
| SO2-fuzzy-standard | Standard (fixed) | Soft generator (fuzzy angles) |
| SO2-fuzzy-general | Learned basis (Stiefel) | Soft generator (fuzzy angles) |

Each codon $c \in \mathcal{C}_a$ is represented by the block-diagonal matrix:

$$D_a^{(i)} = \bigoplus_{j=1}^{d/2} \begin{bmatrix} \cos(\theta_{a,j}^{(i)}) & -\sin(\theta_{a,j}^{(i)}) \\ \sin(\theta_{a,j}^{(i)}) & \cos(\theta_{a,j}^{(i)}) \end{bmatrix}.$$

where $R(\theta) \in SO(2)$ is the standard planar rotation matrix. This generalization retains the equivariant group action structure: the embedding of codon $c$ is obtained by applying the same group label $\varphi_a(c) \in \mathbb{Z}_{n_a}$ to all subspaces, but using separate generators for each.

Because each block acts independently and satisfies the same group multiplication rules as the base case, the full map $\Phi_{\Theta_a}$ defines a homomorphism from $\mathbb{Z}_{n_a}$ into a block-diagonal subgroup of $SO(d)$. A formal proof is provided in Appendix A.2. This construction enables each amino acid's codon group to learn richer, task-specific rotational structure across multiple subspaces, while still preserving group coherence. The learned parameters $\Theta_a$ thus define a smooth, differentiable family of cyclic representations tailored to the downstream task.

## A.4 Equivariance Enforcement via Auxiliary Loss

A central goal of our codon embedding framework is to respect the symmetry structure induced by synonymous codons. Specifically, we aim to ensure that the learned representations are *equivariant* to the group action defined by codon substitutions; that is, rotating a codon within its synonymous group should induce a predictable and structured transformation in the model's internal representation.

While our embedding construction encodes codon-level structure using $SO(2)$ or block-diagonal $SO(d)$ transformations, these symmetries can be diluted or forgotten by subsequent layers in the model if not explicitly preserved. To address this, we introduce an **equivariance regularization mechanism** that reinforces the geometric structure throughout the network by aligning the transformations induced by codon substitutions with transformations in representation space.

**Equivariance objective.** Let $\mathbf{z}_A \in \mathbb{R}^d$ denote the base embedding of amino acid $A$, and let $R_c \in SO(d)$ be the rotation matrix corresponding to codon $c \in \mathcal{C}_A$. The codon-specific embedding is then $\mathbf{E}(c) = R_c \mathbf{z}_A$. To enforce equivariance in the downstream model $f(\cdot)$, we pass both the base embedding $\mathbf{z}_A$ and the rotated codon embedding $\mathbf{E}(c)$ into the network. Specifically, we require that the output of the network behaves consistently under this transformation:

$$f(R_c \mathbf{z}_A) \approx R_c f(\mathbf{z}_A). \tag{2}$$

To promote this behavior, we define an **equivariance loss term**:

$$\mathcal{L}_{\text{equiv}} = \mathbb{E}_{(A,c)} \left[ \|f(R_c \mathbf{z}_A) - R_c f(\mathbf{z}_A)\|^2 \right], \tag{3}$$

which penalizes deviations from equivariance over the distribution of codons $c$ for each amino acid $A$. This term is added to the primary task loss, resulting in a total objective:

$$\mathcal{L}_{\text{total}} = \mathcal{L}_{\text{task}} + \lambda_{\text{equiv}} \mathcal{L}_{\text{equiv}},$$

where $\lambda_{\text{equiv}}$ controls the strength of the symmetry regularization.

Enforcing equivariance yields several advantages. First, it ensures that synonymous codon substitutions, modeled as group actions, produce predictable and structured effects in the model's internal states, enhancing interpretability and robustness. Second, it encourages the model to generalize better across unseen synonymous codons or species with different codon usage, since synonymous substitutions correspond to smooth, equivariant transformations rather than arbitrary shifts in representation. Third, it introduces an inductive bias that acts as a regularizer, improving performance in low-data regimes and enabling transfer to biologically diverse contexts.

Importantly, equivariance also facilitates **fine-tuning of rotation angles** $\theta_a$ in downstream tasks. Since the network is trained to be equivariant with respect to codon group actions, small updates to the generator angles $\theta_a$ (e.g., to adapt to species-specific codon bias or task-specific codon effects) do not destabilize the representation space. Instead, the model can smoothly adjust the codon orbits without breaking its internal alignment, allowing the embedding geometry to shift in a controlled and interpretable manner. This makes the framework especially well-suited for transfer learning scenarios and downstream finetuning tasks where codon preference may shift but the overall group structure remains consistent.

## A.5 Equivariant Pooling Mechanisms for SO(2) Representations

### A.5.1 Cartesian Product of SO(2) Groups

The embedding space in our model consists of $m$ independent 2D subspaces, each encoding a rotation in $SO(2)$. These subspaces correspond to $m$ codon-specific blocks in the full embedding vector of dimension $D = 2m$. Each block lies in $\mathbb{R}^2$ and encodes a codon- or amino-acid-level transformation as a rotation vector $(\cos\theta_i, \sin\theta_i)$ for some angle $\theta_i \in [0, 2\pi)$. To model this structure formally, we consider the total embedding space as the Cartesian product of $m$ copies of the special orthogonal group $SO(2)$:

$$G = \underbrace{SO(2) \times SO(2) \times \cdots \times SO(2)}_{m \text{ times}} = SO(2)^m.$$

This group $G = \mathrm{SO}(2)^m$ is a direct product of Lie groups and forms a smooth manifold equipped with a natural group operation defined component-wise. Each element $g \in G$ can be written as a tuple of rotation matrices:

$$g = (R(\theta_1), R(\theta_2), \ldots, R(\theta_m)),$$

where $R(\theta_i) \in \mathrm{SO}(2)$ is a $2 \times 2$ matrix representing a rotation in the $i$-th subspace:

$$R(\theta_i) = \begin{bmatrix} \cos\theta_i & -\sin\theta_i \\ \sin\theta_i & \cos\theta_i \end{bmatrix}.$$

The group operation in $G$ is defined component-wise:

$$g \cdot h = (R(\theta_1)R(\phi_1), R(\theta_2)R(\phi_2), \ldots, R(\theta_m)R(\phi_m)),$$

for $g = (R(\theta_1), \ldots, R(\theta_m))$ and $h = (R(\phi_1), \ldots, R(\phi_m))$.

This product structure induces a natural notion of independent equivariance in each rotational subspace. That is, for a transformation to be equivariant under $G$, it must commute with the action of each $R(\theta_i)$ individually:

$$f(g \cdot x) = g \cdot f(x), \quad \forall g \in G,\ x \in \mathbb{R}^{2m}.$$

In our model, each codon embedding is represented in this $G$-equivariant space. Consequently, pooling over a sequence of such codon embeddings should ideally respect the product group structure. This justifies using blockwise angle-space aggregation, in which pooling operates independently within each $\mathrm{SO}(2)$ factor.

**Biological Relevance.** The product structure $\mathrm{SO}(2)^m$ captures the modularity inherent in codon composition. Each codon position contributes independently to the semantic representation of the sequence, analogous to how each gene segment contributes to protein folding or expression. Modeling the embedding space as $\mathrm{SO}(2)^m$ allows for biologically meaningful invariances (e.g., synonymous codon substitutions) to be explicitly encoded in the architecture, enhancing robustness and generalization.

**Connection to Equivariant Pooling.** Let $\theta^{(l)} \in \mathbb{R}^m$ denote the angle vector at sequence position $l$ for each of the $m$ rotation blocks. The pooling operation

$$\theta_{\text{pooled}} = \sum_{l=1}^{L} w_l\, \theta^{(l)} \bmod 2\pi$$

defines a mapping from $G^L \to G$ that is a group homomorphism when the weights $w_l$ form a convex combination. This mapping is the formal basis of our $\mathrm{SO}(2)$-aware pooling mechanisms introduced in the next section.

To preserve the equivariant structure of codon embeddings under the action of the group $\mathrm{SO}(2)$, it is crucial to aggregate sequence representations in a way that respects the geometric nature of the embeddings. Standard mean or max pooling in Cartesian space may violate rotational equivariance, as these operations do not align with the circular topology of angle-based embeddings.

Given codon embeddings where each pair of dimensions represents a 2D rotation vector $(x_i, y_i) \in \mathbb{R}^2$ corresponding to a rotation matrix $R(\theta_i) \in \mathrm{SO}(2)$, we aim to construct a pooling function $\phi$ that maps the Cartesian product group $G = G_1 \times \cdots \times G_n$ (where each $G_i \subseteq \mathrm{SO}(2)$) to a single $\mathrm{SO}(2)$ element while preserving group structure.

**Homomorphic Averaging.** Let each codon embedding correspond to a group element $R(\theta_i) \in \mathrm{SO}(2)$, and consider weights $w_i \geq 0$ such that $\sum_{i=1}^{n} w_i = 1$. Define the mapping

$$\phi\Big(R(\theta_1), \ldots, R(\theta_n)\Big) = R\left(\sum_{i=1}^{n} w_i\, \theta_i \bmod 2\pi\right).$$

We can verify that $\phi$ is a homomorphism:

$$\phi(g \cdot h) = \phi(g) \cdot \phi(h), \quad \forall g, h \in G,$$

which ensures that equivariant properties are preserved under pooling. This motivates the use of angle-space aggregation as a principled pooling strategy when working with rotation-encoded representations.

We now describe three such mechanisms:

### A.5.2 Polar Pooling

Polar pooling converts each 2D embedding $(x, y)$ into its polar form $(r, \theta)$, pools the representations in the complex plane using magnitude-weighted averaging, and then converts the result back to Cartesian coordinates. Specifically, for a sequence of vectors $\{(x_i, y_i)\}_{i=1}^{L}$, we compute the complex representation $z_i = r_i e^{i\theta_i}$ and define:

$$z_{\text{pooled}} = \frac{1}{L} \sum_{i=1}^{L} z_i, \quad (x_{\text{pooled}}, y_{\text{pooled}}) = (\Re(z_{\text{pooled}}), \Im(z_{\text{pooled}})).$$

This method preserves the angular structure and is invariant to rotation magnitude noise. It has the advantage of being simple to implement and interpret while respecting the geometry of $SO(2)$.

### A.5.3 DFT-Based Rotation-Aware Pooling

We further introduce a frequency-domain pooling method that performs a Discrete Fourier Transform (DFT) along the sequence length for each $SO(2)$ block. The core idea is to learn a spectral filter $W \in \mathbb{C}^m$ applied in the frequency domain:

$$X = \text{FFT}(\mathbf{x}), \quad X_{\text{filtered}} = X \odot W, \quad \mathbf{x}_{\text{pooled}} = \Re(\text{IFFT}(X_{\text{filtered}})).$$

This allows the model to learn frequency-specific weighting, acting as a soft attention over periodic patterns in the sequence. Additionally, angular modulation via learned sinusoidal positional weights enhances sensitivity to codon position and rhythm, making this approach well-suited for detecting periodic or repeating structures in mRNA.

### A.5.4 SO(2) Mean Pooling

In this approach, we directly pool in angle space by computing the weighted average of the rotation angles. For each codon embedding, we first convert it to angle space via:

$$\theta_{i,j} = \text{atan2}(y_{i,j}, x_{i,j}),$$

and apply attention weights $\alpha_i$ derived from the embedding magnitudes:

$$\theta_{\text{pooled},j} = \text{atan2}\left(\sum_i \alpha_i \sin \theta_{i,j}, \sum_i \alpha_i \cos \theta_{i,j}\right).$$

The final embedding is then reconstructed as:

$$(x_{\text{pooled}}, y_{\text{pooled}}) = (\cos \theta_{\text{pooled}}, \sin \theta_{\text{pooled}}).$$

This pooling strategy is explicitly equivariant under the action of $SO(2)$ and allows interpretable analysis of angular dynamics within a sequence.

**Advantages.** All three pooling mechanisms respect the group structure of $SO(2)$ and support equivariant downstream processing. Compared to standard pooling, these approaches:

- Preserve biologically meaningful angular relationships between synonymous codons.
- Enable learnable or data-driven aggregation of periodic structures in sequence space.
- Provide a theoretically grounded alternative to Cartesian pooling that aligns with group-theoretic design principles.

By incorporating these pooling methods, we enable robust, equivariant representations in our mRNA models, which are crucial for generalization across different codon usage patterns and species.

### A.6 Mathematical Details and Proofs

**Proposition A.1** (Codon Mapping Homomorphism with Arbitrary Generator). *Let $G_a \cong \mathbb{Z}_{n_a}$ be the group of synonymous codon substitutions for amino acid $a$, and let $\varphi_a : \mathcal{C}_a \to \mathbb{Z}_{n_a}$ be a bijective labeling of its codons. Let $\theta_a \in \mathbb{R}$ be any fixed angle such that $n_a \cdot \theta_a \equiv 0 \pmod{2\pi}$. Define the map*

$$\Phi_{\theta_a} : \mathcal{C}_a \to SO(2), \quad \Phi_{\theta_a}(c) = R\big(\varphi_a(c) \cdot \theta_a\big).$$

*Then $\Phi_{\theta_a}$ is a group homomorphism from $G_a$ into $SO(2)$. Moreover, if $\theta_a$ is a generator of a cyclic subgroup of $SO(2)$ of order $n_a$ (i.e., $\theta_a = \frac{2\pi m}{n_a}$ with $\gcd(m, n_a) = 1$), then $\Phi_{\theta_a}$ is an isomorphism onto its image.*

*Proof.* Let $c_1, c_2 \in \mathcal{C}_a$ with integer labels $k_1 = \varphi_a(c_1)$ and $k_2 = \varphi_a(c_2)$. The group operation in $G_a$ corresponds to codon composition

$$c_3 := \varphi_a^{-1}\big((k_1 + k_2) \bmod n_a\big).$$

Then,

$$\Phi_{\theta_a}(c_1)\, \Phi_{\theta_a}(c_2) = R(k_1\theta_a)\, R(k_2\theta_a) = R((k_1 + k_2)\theta_a) = \Phi_{\theta_a}(c_3),$$

by the additive property of rotations. Hence, $\Phi_{\theta_a}$ preserves the group operation and is a homomorphism. To establish injectivity (and thus isomorphism onto its image), note that $R(j\theta_a) = R(j'\theta_a)$ if and only if $(j - j')\theta_a \equiv 0 \pmod{2\pi}$.
This implies $j \equiv j' \pmod{n_a}$ whenever $n_a\theta_a \equiv 0 \pmod{2\pi}$ and $\gcd(m, n_a) = 1$ where $\theta_a = \frac{2\pi m}{n_a}$. In this case, $\Phi_{\theta_a}$ is injective on $\mathbb{Z}_{n_a}$, and hence bijective from $\mathcal{C}_a$ to its image in $SO(2)$.
The image $\Phi_{\theta_a}(\mathcal{C}_a)$ forms a cyclic subgroup of $SO(2)$ of order $n_a$, generated by $R(\theta_a)$. Therefore, $\Phi_{\theta_a}$ is an isomorphism from $G_a$ onto this subgroup. $\qquad\square$

**Proposition A.2** (Cyclic Group Representation via Block-Diagonal Rotations). *Let $G_a \cong \mathbb{Z}_{n_a}$ be the substitution group for amino acid $a$, and let*

$$\Theta_a = (\theta_a^{(1)}, \ldots, \theta_a^{(d/2)}) \in \mathbb{R}^{d/2}$$

*be a set of generator angles. Define the embedding map:*

$$\Phi_{\Theta_a}(j) = blockdiag\big(R(j\theta_a^{(1)}), \ldots, R(j\theta_a^{(d/2)})\big) \in SO(d),$$

*where $R(\cdot) \in SO(2)$, and $j \in \mathbb{Z}_{n_a}$.*
*Then $\Phi_{\Theta_a}$ is a group homomorphism from $\mathbb{Z}_{n_a}$ into $SO(d)$, and its image forms a cyclic subgroup of $SO(d)$ of order dividing $n_a$, with equality if $\gcd(n_a, m_i) = 1$ for all $i$, where $\theta_a^{(i)} = \frac{2\pi m_i}{n_a}$.*

*Proof.* Each subspace map $j \mapsto R(j\theta_a^{(i)})$ is a homomorphism from $\mathbb{Z}_{n_a} \to SO(2)$, and by Proposition A.1, forms a cyclic subgroup of $SO(2)$ of order dividing $n_a$. Since block-diagonal matrix multiplication distributes over block components, the full map $\Phi_{\Theta_a}(j_1 + j_2) = \Phi_{\Theta_a}(j_1)\Phi_{\Theta_a}(j_2)$, proving that $\Phi_{\Theta_a}$ is a homomorphism. Injectivity (i.e., full cyclic order $n_a$) holds if each $\theta_a^{(i)} = \frac{2\pi m_i}{n_a}$ with $\gcd(m_i, n_a) = 1$, ensuring each component cycles with full order. Thus the image of $\Phi_{\Theta_a}$ is a cyclic subgroup of $SO(d)$. $\qquad\square$

**Proposition A.3** (Codon Embeddings via Stiefel-Manifold Subspaces). *Let $V_A \in St(2, d)$ be an orthonormal basis for a 2D subspace of $\mathbb{R}^d$, and let $\mathbf{z}_A \in \mathbb{R}^d$ be the base vector for amino acid $A$. Define $\mathbf{E}(c) := V_A R(\phi_c) V_A^\top \mathbf{z}_A$, where $R(\phi_c) \in SO(2)$. Then:*

1. *$\mathbf{E}(c) \in \mathbb{R}^d$ lies on a circle in the subspace spanned by $V_A$.*

2. *The transformation $V_A R(\phi_c) V_A^\top \in SO(d)$ acts as a rotation in the 2D subspace and as identity on the orthogonal complement.*

*Proof.* (1) Since $V_A^\top \mathbf{z}_A \in \mathbb{R}^2$, the codon embedding $\mathbf{E}(c) = V_A R(\phi_c) V_A^\top \mathbf{z}_A$ is the image of a 2D rotation applied to the projection of $\mathbf{z}_A$ into the subspace defined by $V_A$. As $R(\phi_c)$ preserves Euclidean norm, all embeddings lie at radius $\|V_A^\top \mathbf{z}_A\|$, tracing a circle.

(2) The matrix $V_A R(\phi_c) V_A^\top$ is a similarity transformation acting only in the 2D plane spanned by $V_A$. It is orthogonal because:

$$(V_A R V_A^\top)(V_A R V_A^\top)^\top = V_A R R^\top V_A^\top = V_A V_A^\top,$$

which is a projection matrix onto the 2D subspace. Within that subspace, the operator behaves exactly as $R(\phi_c) \in SO(2)$. On the orthogonal complement, it acts as identity since the projection of $\mathbf{z}_A$ is zero. Hence the overall transformation is a block rotation in $SO(d)$. □

**Proposition A.4** (Sequence Homomorphism). *Let's denote our mapping as*

$$\phi\Big(R(\theta_1), R(\theta_2), \ldots, R(\theta_n)\Big) = R\Big(\sum_{i=1}^{n} w_i\, \theta_i \mod 2\pi\Big),$$

*where each $R(\theta_i)$ is a rotation in a subgroup of SO(2) and $w_i$ are fixed (or learned) weights. We want to show that $\phi$ is a homomorphism from the Cartesian product group*

$$G = G_1 \times G_2 \times \cdots \times G_n$$

*into $H \subset SO(2)$. In other words, for any two elements*

$$g = \big(R(\alpha_1), R(\alpha_2), \ldots, R(\alpha_n)\big)$$

*and*

$$h = \big(R(\beta_1), R(\beta_2), \ldots, R(\beta_n)\big)$$

*in $G$, we need to prove that*

$$\phi(g \cdot h) = \phi(g) \cdot \phi(h).$$

*Proof.* Group Operation in $G$: Since each $G_i$ is a subgroup of SO(2), its group operation is given by addition of angles (modulo $2\pi$). Thus, the group operation in $G$ is defined coordinate-wise:

$$g \cdot h = \Big(R(\alpha_1 + \beta_1),\, R(\alpha_2 + \beta_2),\, \ldots,\, R(\alpha_n + \beta_n)\Big).$$

Using the definition of $\phi$, we have

$$\phi(g \cdot h) = \phi\Big(R(\alpha_1 + \beta_1),\, \ldots,\, R(\alpha_n + \beta_n)\Big) = R\left(\sum_{i=1}^{n} w_i\,(\alpha_i + \beta_i) \mod 2\pi\right).$$

Because the weighted sum is linear, we can write

$$\sum_{i=1}^{n} w_i\,(\alpha_i + \beta_i) = \sum_{i=1}^{n} w_i\,\alpha_i + \sum_{i=1}^{n} w_i\,\beta_i.$$

Therefore,

$$\phi(g \cdot h) = R\left(\sum_{i=1}^{n} w_i\,\alpha_i + \sum_{i=1}^{n} w_i\,\beta_i \mod 2\pi\right).$$

$$R(a) \cdot R(b) = R(a + b \mod 2\pi).$$

Thus, we have

$$R\left(\sum_{i=1}^{n} w_i\,\alpha_i + \sum_{i=1}^{n} w_i\,\beta_i\right) = R\left(\sum_{i=1}^{n} w_i\,\alpha_i\right) \cdot R\left(\sum_{i=1}^{n} w_i\,\beta_i\right).$$

By the definition of $\phi$,

$$\phi(g) = R\left(\sum_{i=1}^{n} w_i\,\alpha_i\right), \quad \phi(h) = R\left(\sum_{i=1}^{n} w_i\,\beta_i\right).$$

Therefore, we obtain

$$\phi(g \cdot h) = \phi(g) \cdot \phi(h).$$

$\square$

## A.7 Background

Advances in genomic language modeling have introduced a range of strategies for representing biological sequences, each with different trade-offs in terms of biological fidelity, expressiveness, and computational efficiency. At the core of this challenge lies the question of whether models should operate on raw nucleotide sequences, protein-level amino acids, or intermediate representations such as codons. Each level captures distinct aspects of the biological signal: nucleotide-level models retain the full sequence information, amino acid models emphasize protein function, and codon-level models provide a balance by preserving the coding frame and capturing synonymous variation. Understanding how these representations influence model performance is critical for designing language models that are both biologically grounded and effective across a wide range of downstream tasks.

### A.7.1  Nucleotide-level models (DNA/RNA as a sequence of bases)

Nucleotide-level tokenization treats the coding sequence simply as a long sequence of nucleotides (characters A, C, G, T/U). Traditional sequence models like recurrent neural networks or transformers can be applied at the base level, sometimes using sliding windows or overlapping k-mers for tractability. For example, early genomic LMs such as DNABERT tokenized DNA sequences into overlapping k-mers (e.g. 6-mers) and then learned representations with the BERT architecture [20]. Nucleotide-level models have the advantage of retaining all information (including non-coding regions or regulatory motifs in the sequence context), but they do not inherently recognize the codon structure. Synonymous codons will be treated as completely unrelated tokens unless the model implicitly learns their interchangeability from data. The vocabulary (4 nucleotides or a set of k-mers) is relatively small, but sequences are three times longer at the nucleotide level than at the codon level, which can pose challenges for model input length and learning long-range dependencies. Models like DNABERT [20] and the Nucleotide Transformer [8] tokenize sequences into overlapping $k$-mers (e.g., 6-mers), striking a balance between biological context and computational tractability. While effective at capturing regulatory signals and enabling genome-scale analysis, these models lack an inherent inductive bias toward codon structure unless explicitly modeled. Recent approaches, such as DNABERT-2 [55] and GROVER [41], explore adaptive tokenization methods like byte-pair encoding to better reflect sequence regularities. Collectively, these models have demonstrated strong performance across a variety of genomics tasks by leveraging patterns learned directly from nucleotide-level input [2, 54].

### A.7.2  Amino acid-level models (operate on the translated protein sequence)

Here, the DNA coding sequence is first translated to the corresponding amino acid sequence, and then a protein language model is applied. This effectively discards the specific codon information, reducing the sequence to the 20-standard amino acid alphabet. Many state-of-the-art protein language models use this approach, treating proteins as sequences of amino acids (e.g., ESM and related transformer models [24, 38, 11]). By focusing on amino acids, these models tap into signals of protein structure/function and avoid the complication of extremely long nucleotide sequences. Indeed, large protein LMs have shown remarkable success in predicting structure and function directly from

sequence embeddings [38]. However, by design amino-acid models ignore synonymous codon variation. They cannot capture codon bias effects on translation or gene expression, since all synonymous DNA sequences collapse into the same amino acid sequence. For tasks where the protein phenotype is paramount (e.g. structural modeling), this may be acceptable [38], but for tasks related to gene expression, regulatory control, or species-specific usage, amino-acid–only models miss a vital piece of information. As an example, codon usage biases differ by species, and aspects like translation elongation rates or mRNA stability are influenced by the specific codon sequence not just the resulting protein. Thus, while amino acid LMs leverage evolutionary signals in protein space, they lack sensitivity to codon-level features by construction. Synonymous codon usage has been linked to specific structural characteristics of proteins [42, 39], and accumulating evidence highlights a relationship between codon usage patterns and protein folding processes [30, 26]. Such relationships suggest that codon-level signals may carry essential structural information currently missed by popular protein structure predictors, such as ESMfold [24] and OmegaFold [50], which predominantly leverage language models designed to identify sequence-level correlations rather than explicit physical principles of folding. Given that existing deep learning approaches have been shown to inadequately capture the underlying biophysical mechanisms governing protein folding [32], incorporating synonymous codon usage signals could significantly improve structural predictions.

**Codon-level or 3-mer models (using codons as tokens)**    A compromise approach is to tokenize the sequence in codons (triplets of nucleotides) or other 3-mer genomic words, so that each token represents a genomic substring of length three. In coding regions, aligning 3-mers precisely to the reading frame is particularly intuitive, as each token directly corresponds to one amino acid (or a stop signal) in the protein sequence. This representation preserves synonymous codon distinctions while simultaneously reducing sequence length by three-fold compared to nucleotide-level models. The resulting vocabulary of codons includes 64 tokens (61 sense codons plus 3 stop signals), a size somewhat larger than the amino acid alphabet but still highly practical for language modeling. Recent approaches such as CodonBERT [23], HELM [52], CaLM [36], and DNABERT using 6-mers [56] have shown that codon-level representations can outperform standard amino acid models by capturing nuanced codon bias signals like species-specific preferences and rare codon placements [31, 32]. Overall, encoding sequences at the codon level represents a straightforward yet powerful strategy for embedding explicit knowledge of the coding frame into language models.

Codon-level tokenization captures synonymous differences but fails to reflect the biological relationships between codons that encode the same amino acid. Hierarchical approaches, such as HELM (Hierarchical Encoding for mRNA Language Modeling) [52], address this by explicitly modeling the codon-amino acid structure during pretraining. HELM introduces a codon hierarchy into the loss function, penalizing the model less for predicting synonymous codons, thus encouraging embeddings that cluster biologically equivalent codons together. This alignment with the genetic code significantly improves both predictive and generative performance, outperforming standard models by  8% across diverse tasks and producing sequences that better reflect natural codon usage. These results highlight the value of embedding biologically grounded inductive biases into language models for mRNA.

### A.7.3    Limitations of Codon-Level Embedding Models and Motivation for Group-Theoretic Embeddings

Despite their advantages, codon-level embedding language models have important limitations that constrain their robustness and generalizability. A critical drawback is that the learned codon embeddings typically lack interpretable or meaningful geometric relationships: although synonymous codons may cluster in the embedding space due to the structure of the loss function, these models do not explicitly encode biologically meaningful relationships among codons. For instance, while HELM [52] implicitly prefers synonymous codons due to optimization pressures, it does so without establishing tangible structural or biological relationships between codons in the embedding space. As a consequence, the embeddings learned by such models can be brittle; small perturbations in the embedding vectors or minor deviations from the training distribution may cause significant degradations in model performance or even nonsensical outputs.

Another critical limitation stems from the difficulty of adapting these embeddings to new species or contexts via fine-tuning. Codon usage preferences vary significantly across organisms [35], and ideally, a model should easily adjust its embeddings to capture species-specific codon biases. However, current codon-level embedding methods are typically not robust to fine-tuning, as the

entire downstream network heavily relies on the original embedding structure. Small adjustments in codon embedding vectors, intended to reflect new codon preferences, often propagate unpredictably throughout the network, effectively destabilizing the representations and yielding outputs that degrade biological fidelity or model accuracy. Thus, rather than enabling straightforward adaptation, attempts at fine-tuning may inadvertently disrupt the learned codon relationships, resulting in a substantial loss of performance and biological interpretability.

### A.7.4 Equivariant Architectures and Symmetry-Aware Modeling

Incorporating symmetry into neural architectures has emerged as a powerful inductive bias for learning over structured domains. Group equivariant neural networks (G-CNNs) extend the notion of translational equivariance in convolutional networks to more general symmetry groups, enabling the model to respect intrinsic invariances in the data [21, 7, 3]. While early applications of equivariance were rooted in image domains, exploiting rotation and reflection symmetries, recent works have expanded these ideas into non-Euclidean settings and non-linear operators such as self-attention. For instance, the LieTransformer [17] generalizes attention mechanisms to arbitrary Lie groups, demonstrating equivariant self-attention in diverse domains including molecular property prediction and physical trajectory modeling. Similarly, TorchMD-Net [47] highlights the advantage of rotationally equivariant transformers in modeling vectorial and tensorial outputs for molecular systems, showing improved generalization even for scalar-valued properties.

Rotationally equivariant neural networks have proven especially effective in domains where predicted outputs are vectors or tensors. SE(3)-Transformers [14], Cormorant [1], and equivariant message passing networks [43] leverage geometric symmetry to model 3D molecular systems with higher accuracy and efficiency. While these models were developed outside of biology, they highlight the broader value of enforcing structured equivariance. Inspired by these principles, our approach introduces a form of rotational symmetry within the codon space of mRNA, aiming to reduce redundancy among synonymous codons through group-theoretic modeling.

To address these shortcomings, there is strong motivation to develop embedding frameworks that explicitly integrate the known symmetry and redundancy of the genetic code from the outset. Incorporating codon synonym groups as mathematically defined equivalence classes could substantially reduce model complexity and allow models to directly exploit biologically relevant variation, such as species-specific codon usage patterns, without the risk of destabilizing learned representations. This approach aligns closely with established biological knowledge and promises scientifically-grounded AI models that are inherently data-efficient, interpretable, and robust. Ultimately, creating such structured codon representations would not only address the current limitations of existing language modeling techniques for DNA and RNA but would also enable scalable AI systems capable of effectively navigating vast genomic sequence spaces for critical health-related applications, including predicting gene expression or designing optimized mRNAs for vaccines [53].

### A.8 Interpretability Analysis

### A.8.1 Evaluation of Model Interpretability

To rigorously assess whether our SO(2)-structured codon embeddings encode biologically meaningful signals, we performed two targeted analyses.

**1. GC Content vs. Entropy of Codon Angles**   Genomic GC content is a major driver of codon usage bias [34]. We therefore measured how dispersed the learned codon angles are as a function of transcript GC%. For each coding sequence $s$ of length $N$ codons, let $\theta_1, \ldots, \theta_N \in [0, 2\pi)$ denote the model's predicted angles. We estimate the empirical distribution

$$p_i = \frac{1}{N} \sum_{j=1}^{N} \mathbf{1}(\theta_j \in \mathrm{bin}_i), \quad i = 1, \ldots, K$$

by binning the circle into $K$ equal-width intervals. The Shannon entropy of the angle distribution is then

$$H(s) = -\sum_{i=1}^{K} p_i \log_2 p_i.$$

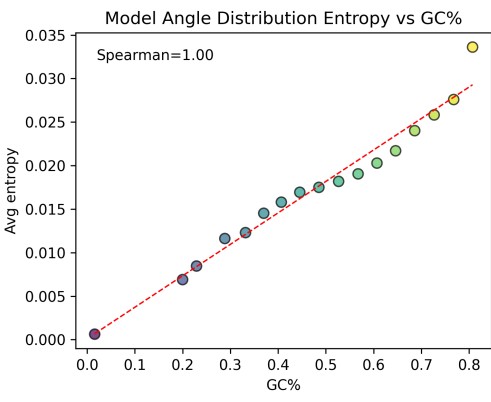
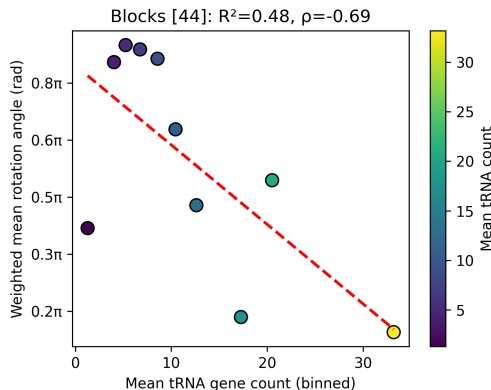

| (a) Angle Distribution Entropy vs. GC-Content | (b) Weighted Mean Angle vs. tRNA Abundance |
|---|---|

Figure 5: **Interpretability of SO(2) codon embeddings.** (a) Mean Shannon entropy of learned codon angles, binned by transcript GC-content, exhibits a near-linear increase (Pearson $r = 0.98$, $R^2 = 0.97$, $p < 10^{-11}$), confirming that GC-driven biases are captured in angular dispersion. (b) Weighted mean rotation angle per codon versus normalized human tRNA gene copy number (top correlating block), showing that codons decoded by more abundant tRNAs are assigned systematically smaller angles (Spearman $\rho = -0.69$, $p < 10^{-6}$), indicating the embedding's alignment with translational efficiency.

We grouped all human transcripts into $M$ GC% bins (e.g. 20 equal-width bins on $\text{GC}\% \in [0, 1]$) and computed the mean entropy $\overline{H}$ in each bin. Fitting a linear model

$$\overline{H} = \alpha + \beta \, \text{GC}\% + \varepsilon$$

yielded $\beta = 0.35 \pm 0.01$ (SE), Pearson $r = 0.98$, $R^2 = 0.97$, $p < 10^{-11}$. This near-perfect trend demonstrates that higher GC content known to restrict codon choice to G/C-rich codons corresponds to lower angle uncertainty, validating that our embeddings capture mutational biases [44, 34].

**2. tRNA Abundance vs. Codon Angle** Translational efficiency is strongly influenced by tRNA gene copy number [19]. For each of the 61 sense codons $c$, let $\theta_c$ be its learned angle (anchored so $\theta_c \in [0, 2\pi)$), and let $t_c$ be its normalized tRNA count. We assessed monotonic association via Spearman's rank correlation:

$$\rho = 1 - \frac{6 \sum_c (r(\theta_c) - r(t_c))^2}{61 \, (61^2 - 1)},$$

where $r(\cdot)$ denotes the rank. We observed $\rho = -0.69$ ($p < 10^{-6}$), indicating that codons with more abundant tRNAs are systematically placed at smaller angles. This alignment shows that the SO(2) embedding axis faithfully recapitulates the biological continuum of "optimal" to "non-optimal" codons [19, 45].

**Significance and Interpretation** These results carry three key implications:

1. *Biological alignment of latent space.* The strong GC–entropy trend (Pearson $r = 0.98$) confirms that the embedding's angular dispersion encodes mutational biases in codon usage.

2. *Translational efficiency signal.* The negative Spearman $\rho = -0.69$ substantiates that the continuous angular coordinate corresponds to tRNA-driven translation speed, without explicit supervision.

3. *Enhanced interpretability.* Constraining codon embeddings to a one-dimensional circle yields a latent space whose every point has direct biological meaning—mutational constraint or translational optimality thereby increasing model transparency and trust.

Overall, these analyses justify our use of an SO(2) prior: it does not limit expressivity, but rather guides the model to learn the dominant forces shaping codon usage, producing a compact, interpretable representation amenable to downstream biological insights.

### A.9 Experimental Setup

#### A.9.1 Pretraining Corpus Construction

All RefSeq files with the suffix ".rna.gbff.gz" were downloaded via NCBI's FTP server. Using the Helical AI curation toolkit [49], we parsed each GenBank record, extracted 5'-UTR, CDS, and 3'-UTR features, and retained only the CDS regions to ensure that our embeddings capture codon-level translational signals rather than untranslated flanking sequences. Sequences shorter than 20 codons or longer than 512 codons, or containing non-canonical bases, were discarded.

Due to computational limits, we randomly subsampled the cleaned set of approximately 56 million CDS entries down to 25 million, maintaining the following class proportions: 37.6% other vertebrates, 24.4% mammals, 22.8% invertebrates, 13.7% fungi, and 1.4% viruses (including major human pathogens such as SARS-CoV-2, influenza, RSV, HIV-1/2, HBV, HCV, HSV, EBV, VZV, Zika, and Dengue 1–4). For the low-data ablation, we sampled 1 million sequences from this pool with stratified sampling to preserve these ratios.

All retained coding sequences were then "codonized" by grouping every three nucleotides into a single token, yielding a vocabulary of 64 codons. This pipeline, which includes data acquisition, filtering, codonization, and subsampling, ensures both biological relevance and computational tractability for our SO(2)-based embedding experiments.

#### A.9.2 Downstream Evaluation

We evaluate our models on diverse biologically relevant datasets:

- MLOS (Flu Vaccine mRNAs): Contains 543 sequences of influenza vaccine mRNAs developed by Sanofi-Aventis. The task focuses on predicting the expression levels of these sequences, each approximately 1700 nucleotides long [23].

- mRFP Expression: Includes 1459 synthetic mRNA sequences encoding a red fluorescent protein (mRFP). Each sequence has a fixed length of 678 nucleotides, and the goal is to predict their expression levels quantitatively[29].

- E. coli Protein Expression: Consists of 6348 sequences from E. coli, varying from 171 to 3000 nucleotides. The classification task aims to determine whether a given mRNA sequence results in high or low protein expression.[10]

- Tc-Riboswitches: Comprises 355 riboswitch sequences ranging from 67 to 73 nucleotides. The regression task involves predicting the switching factor, a quantitative measure indicating regulatory activity [15].

- iCodon Human mRNA Stability: A large-scale dataset of 41,123 human mRNA sequences with lengths ranging between 30 and 1497 nucleotides. The objective is to predict mRNA stability as a continuous value [9].

- SARS-CoV-2 Vaccine Degradation: Contains 2400 sequences from SARS-CoV-2 vaccine candidates, each precisely 107 nucleotides. This regression task targets prediction of the degradation rate of vaccine mRNA molecules [48]

All datasets are split consistently into training, validation, and test subsets at ratios of 70%, 15%, and 15%, respectively, for model training and evaluation.

### A.10 Hyperparameters

#### A.10.1 Pretraining Hyperparameters

All twelve SO(2) embedding variants were pretrained under an identical configuration to ensure a fair comparison of their representational effects. We used a 1 M-sequence subset for ablation, with a stratified 25 M corpus reserved for final scaling experiments. Training was run on a single node with eight NVIDIA GPUs, using a per-device batch size of 64 on 8 GPUs and gradient accumulation over 2 steps (effective batch = 1024). Models were trained for 50 epochs, with a 5-epoch linear warmup and cosine lr scheduler. Sequences were truncated or padded to a maximum length of 512 codons. We logged all runs to Weights&Biases and saved checkpoints to the specified output directory. Inputs

and defaults not overridden on the command line (e.g., learning rate, optimizer, scheduler) were left at their parser-specified values.

Table 4 summarizes the key pretraining hyperparameters.

Table 4: Pretraining hyperparameters (identical for all 12 variants).

| Hyperparameter | Value |
| --- | --- |
| Training file | 1M Sample |
| Batch size | 1024 |
| Gradient accumulation steps | 2 |
| Epochs | 50 |
| Warmup epochs | 5 |
| Scheduler | Cosine |
| Max sequence length | 512 codons |
| Precision | bf16-mixed |
| Embedding-layer variants | 12 SO(2) configurations (fixed/cyclic/fuzzy × standard/general) |

Table 5: Pretraining hyperparameters for the 25 M-sequence corpus (identical for both GPT-2 and Mamba hybrid architecture).

| Hyperparameter | Value |
| --- | --- |
| Training file | 25 M Sample |
| Batch size | 1024 |
| Gradient accumulation steps | 2 |
| Epochs | 20 |
| Warmup epochs | 2 |
| Scheduler | Cosine |
| Max sequence length | 512 codons |
| Precision | bf16-mixed |
| Embedding-layer variants | SO2-fuzzy-general-true |

**Downstream Hyperparameter Optimization**   To tune our downstream heads for both generative and property-prediction tasks, we performed a structured grid search over key training parameters. We varied batch size $\in \{8, 16, 32, 64\}$, learning rate $\in \{1e{-}4, 1e{-}5\}$, hidden-ratio $\in \{2, 4, 8\}$ (controls MLP width), and number of head layers $\in \{2, 4, 7\}$. We also swept over two finetuning regimes angle-only and last-two-layers—and five pooling strategies (`so2_mean`, `dft`, `polar`, `mean`, `lie_avg`). Experiments used a fixed random seed (42), early stopping patience of 20 epochs, and scheduler patience of 10 on validation Spearman (regression) or accuracy (classification). Invalid combinations (e.g., angle finetuning on fixed-$\theta$ models) were automatically pruned. The full search space and the selected best hyperparameters for each dataset are reported in Appendix A.10.2.

### A.10.2   Best Downstream Hyperparameters

To provide clarity on our tuning process, we present five tables (Tables 6 through 12) that summarize the single best hyperparameter configuration—across batch size, learning rate, MLP depth and width, finetuning regime, and pooling method—for each downstream dataset. Each table lists the model variant (fixed, cyclic, or fuzzy), whether a Stiefel basis or equivariance loss was used, and the corresponding "Angle?" and "Last2?" flags. By consolidating these results, readers can directly see which configuration achieved the highest validation performance on each task without exhaustive per-dataset details.

Table 6: Downstream hyperparameter configurations for `COV Deg`.

| Model | Stiefel | Equiv. | #Layers | Batch | LR | Angle? | Last2? | Hidden | Pooling |
|---|---|---|---|---|---|---|---|---|---|
| Vanilla | - | - | 4 | 8 | 0.0001 | - | - | 8 | mean |
| Fixed $\theta$ | ✗ | ✗ | 4 | 16 | 0.0001 | ✗ | ✓ | 4 | so2_mean |
| Fixed $\theta$ | ✗ | ✓ | 4 | 32 | 0.0001 | ✗ | ✗ | 4 | polar |
| Fixed $\theta$ | ✓ | ✗ | 4 | 32 | 0.0001 | ✗ | ✓ | 4 | lie_avg |
| Fixed $\theta$ | ✓ | ✓ | 4 | 16 | 1e-05 | ✗ | ✓ | 4 | so2_mean |
| Learned $\theta$ | ✗ | ✗ | 4 | 32 | 0.0001 | ✓ | ✓ | 4 | so2_mean |
| Learned $\theta$ | ✗ | ✓ | 2 | 16 | 1e-05 | ✓ | ✓ | 2 | so2_mean |
| Learned $\theta$ | ✓ | ✗ | 4 | 16 | 1e-05 | ✓ | ✓ | 2 | so2_mean |
| Learned $\theta$ | ✓ | ✓ | 4 | 32 | 0.0001 | ✓ | ✓ | 2 | mean |
| Fuzzy $\theta$ | ✗ | ✗ | 2 | 32 | 0.0001 | ✓ | ✓ | 4 | mean |
| Fuzzy $\theta$ | ✗ | ✓ | 2 | 16 | 0.0001 | ✓ | ✓ | 4 | so2_mean |
| Fuzzy $\theta$ | ✓ | ✗ | 4 | 32 | 0.0001 | ✓ | ✓ | 4 | polar |
| Fuzzy $\theta$ | ✓ | ✓ | 2 | 32 | 1e-05 | ✗ | ✓ | 2 | dft |

Table 7: Downstream hyperparameter configurations for `E. coli`.

| Model | Stiefel | Equiv. | #Layers | Batch | LR | Angle? | Last2? | Hidden | Pooling |
|---|---|---|---|---|---|---|---|---|---|
| Vanilla | - | - | 2 | 8 | 1e-05 | - | - | 4 | mean |
| Fixed $\theta$ | ✗ | ✗ | 2 | 8 | 0.0001 | ✗ | ✓ | 4 | mean |
| Fixed $\theta$ | ✗ | ✓ | 4 | 16 | 1e-05 | ✗ | ✓ | 2 | dft |
| Fixed $\theta$ | ✓ | ✗ | 2 | 8 | 1e-05 | ✗ | ✓ | 4 | so2_mean |
| Fixed $\theta$ | ✓ | ✓ | 4 | 32 | 0.0001 | ✗ | ✓ | 2 | polar |
| Learned $\theta$ | ✗ | ✗ | 4 | 8 | 1e-05 | ✗ | ✓ | 2 | so2_mean |
| Learned $\theta$ | ✗ | ✓ | 2 | 8 | 0.0001 | ✗ | ✓ | 2 | polar |
| Learned $\theta$ | ✓ | ✗ | 4 | 8 | 1e-05 | ✓ | ✓ | 4 | lie_avg |
| Learned $\theta$ | ✓ | ✓ | 2 | 8 | 1e-05 | ✓ | ✓ | 4 | mean |
| Fuzzy $\theta$ | ✗ | ✗ | 4 | 8 | 1e-05 | ✓ | ✓ | 4 | mean |
| Fuzzy $\theta$ | ✗ | ✓ | 2 | 8 | 1e-05 | ✓ | ✓ | 2 | dft |
| Fuzzy $\theta$ | ✓ | ✗ | 2 | 8 | 0.0001 | ✗ | ✓ | 4 | polar |
| Fuzzy $\theta$ | ✓ | ✓ | 2 | 8 | 1e-05 | ✓ | ✓ | 4 | so2_mean |

## A.11 Training Efficiency

To assess the computational overhead introduced by our symmetry-aware embeddings, we measured end-to-end wall-clock training times on both the 1M sequence subset and the full 25M sequence corpus. All models share the same 15M parameter Transformer backbone; variations arise only from the SO(2) embedding configuration (fixed vs. learned vs. fuzzy), choice of basis (standard vs. Stiefel), and inclusion of the equivariance loss.

Table 13 summarizes these results. On the 1M subset, the vanilla model completes in under 2 h, whereas introducing a fixed $\theta$ embedding adds only a modest 30 % overhead when no equivariance loss is used, and roughly doubles training time when the loss is activated. Learned $\theta$ embeddings incur a further cost up to 4–5 $\times$ slower than fixed, reflecting the extra parameterization needed for angle optimization. Stiefel-basis rotations amplify this effect, particularly once the equivariance penalty is applied. Fuzzy $\theta$ variants exhibit the highest training times across all 1 M experiments, as they combine a learned angle distribution with Riemannian updates on the Stiefel manifold.

Scaling to the full 25M corpus magnifies these trends: the hybrid mamba-transformer backbone with fuzzy $\theta$ + Stiefel + equivariance completes in just over one day, while the GPT-2 backbone requires nearly four days under the same configuration. These results highlight a clear trade-off between representational power and compute cost, guiding practical choices for large-scale pretraining under resource constraints.

Table 8: Downstream hyperparameter configurations for `MLOS Split Seed 0`.

| Model | Stiefel | Equiv. | #Layers | Batch | LR | Angle? | Last2? | Hidden | Pooling |
|---|---|---|---|---|---|---|---|---|---|
| Vanilla | - | - | 2 | 32 | 1e-05 | - | - | 2 | mean |
| Fixed $\theta$ | ✗ | ✗ | 4 | 64 | 1e-05 | ✗ | ✗ | 4 | mean |
| Fixed $\theta$ | ✗ | ✓ | 2 | 64 | 0.0001 | ✗ | ✗ | 4 | so2_mean |
| Fixed $\theta$ | ✓ | ✗ | 4 | 32 | 1e-05 | ✗ | ✗ | 2 | lie_avg |
| Fixed $\theta$ | ✓ | ✓ | 2 | 32 | 1e-05 | ✗ | ✓ | 4 | lie_avg |
| Learned $\theta$ | ✗ | ✗ | 8 | 64 | 0.0001 | ✗ | ✓ | 2 | polar |
| Learned $\theta$ | ✗ | ✓ | 2 | 64 | 0.0001 | ✓ | ✓ | 2 | polar |
| Learned $\theta$ | ✓ | ✗ | 2 | 8 | 0.0001 | ✓ | ✓ | 4 | dft |
| Learned $\theta$ | ✓ | ✓ | 4 | 32 | 0.0001 | ✓ | ✗ | 4 | polar |
| Fuzzy $\theta$ | ✗ | ✗ | 2 | 64 | 0.0001 | ✓ | ✓ | 2 | so2_mean |
| Fuzzy $\theta$ | ✗ | ✓ | 4 | 32 | 1e-05 | ✓ | ✗ | 2 | polar |
| Fuzzy $\theta$ | ✓ | ✗ | 8 | 64 | 1e-05 | ✗ | ✓ | 2 | lie_avg |
| Fuzzy $\theta$ | ✓ | ✓ | 4 | 64 | 1e-05 | ✗ | ✓ | 2 | lie_avg |

Table 9: Downstream hyperparameter configurations for `MLOS Split Seed 2`.

| Model | Stiefel | Equiv. | #Layers | Batch | LR | Angle? | Last2? | Hidden | Pooling |
|---|---|---|---|---|---|---|---|---|---|
| Vanilla | - | - | 2 | 32 | 0.0001 | - | - | 8 | mean |
| Fixed $\theta$ | ✗ | ✗ | 4 | 32 | 0.0001 | ✗ | ✓ | 4 | dft |
| Fixed $\theta$ | ✗ | ✓ | 4 | 32 | 0.0001 | ✗ | ✓ | 8 | mean |
| Fixed $\theta$ | ✓ | ✗ | 4 | 32 | 0.0001 | ✗ | ✗ | 4 | lie_avg |
| Fixed $\theta$ | ✓ | ✓ | 2 | 16 | 0.0001 | ✗ | ✓ | 4 | so2_mean |
| Learned $\theta$ | ✗ | ✗ | 2 | 64 | 0.0001 | ✓ | ✗ | 4 | mean |
| Learned $\theta$ | ✗ | ✓ | 2 | 32 | 1e-05 | ✗ | ✗ | 4 | mean |
| Learned $\theta$ | ✓ | ✗ | 4 | 32 | 0.0001 | ✓ | ✗ | 4 | so2_mean |
| Learned $\theta$ | ✓ | ✓ | 4 | 32 | 0.0001 | ✗ | ✓ | 2 | lie_avg |
| Fuzzy $\theta$ | ✗ | ✗ | 2 | 32 | 0.0001 | ✗ | ✗ | 4 | dft |
| Fuzzy $\theta$ | ✗ | ✓ | 2 | 32 | 0.0001 | ✓ | ✓ | 4 | so2_mean |
| Fuzzy $\theta$ | ✓ | ✗ | 2 | 32 | 0.0001 | ✓ | ✗ | 2 | mean |
| Fuzzy $\theta$ | ✓ | ✓ | 4 | 32 | 1e-05 | ✗ | ✗ | 2 | mean |

## A.12 Ablation Study

To evaluate the effect of embedding design on predictive performance, we report results across three correlation-based metrics commonly used in biological sequence modeling: Spearman, Pearson, and coefficient of determination ($R^2$). As shown in Tables 14, 15, and 16.

## A.13 Generation Experiments

This section presents the complete results of our generative evaluation, expanding upon the summary metrics reported in the main text. Table 17 reports Fréchet BioDistance (FBD) scores and generation diversity (IntDiv$_{\text{Gen}}$) across SO(2)-based variants and sampling temperatures. Table 18 provides mean squared error (MSE) of predicted properties for generated suffixes at temperature 1.0. These results offer a comprehensive view of fidelity and diversity trade-offs across embedding configurations and generation settings.

### A.13.1 Fidelity

Table 17 reports Fréchet BioDistance (FBD) scores and generation diversity (IntDiv$_{\text{Gen}}$) across SO(2)-based variants and sampling temperatures.

Table 10: Downstream hyperparameter configurations for `MLOS Split Seed 42`.

| Model | Stiefel | Equiv. | #Layers | Batch | LR | Angle? | Last2? | Hidden | Pooling |
|---|---|---|---|---|---|---|---|---|---|
| Vanilla | - | - | 2 | 16 | 0.0001 | - | - | 4 | mean |
| Fixed $\theta$ | ✗ | ✗ | 4 | 32 | 0.0001 | ✗ | ✓ | 4 | dft |
| Fixed $\theta$ | ✗ | ✓ | 2 | 8 | 1e-05 | ✗ | ✓ | 4 | so2_mean |
| Fixed $\theta$ | ✓ | ✗ | 4 | 16 | 0.0001 | ✗ | ✗ | 2 | dft |
| Fixed $\theta$ | ✓ | ✓ | 4 | 8 | 0.0001 | ✗ | ✓ | 2 | dft |
| Learned $\theta$ | ✗ | ✗ | 4 | 16 | 0.0001 | ✗ | ✓ | 2 | dft |
| Learned $\theta$ | ✗ | ✓ | 2 | 8 | 0.0001 | ✓ | ✓ | 4 | dft |
| Learned $\theta$ | ✓ | ✗ | 4 | 8 | 1e-05 | ✓ | ✓ | 2 | dft |
| Learned $\theta$ | ✓ | ✓ | 2 | 8 | 1e-05 | ✓ | ✗ | 4 | so2_mean |
| Fuzzy $\theta$ | ✗ | ✗ | 4 | 16 | 1e-05 | ✗ | ✓ | 4 | dft |
| Fuzzy $\theta$ | ✗ | ✓ | 2 | 8 | 0.0001 | ✓ | ✓ | 2 | dft |
| Fuzzy $\theta$ | ✓ | ✗ | 2 | 8 | 0.0001 | ✓ | ✗ | 2 | dft |
| Fuzzy $\theta$ | ✓ | ✓ | 2 | 8 | 0.0001 | ✓ | ✓ | 2 | dft |

Table 11: Downstream hyperparameter configurations for `mRFP`.

| Model | Stiefel | Equiv. | #Layers | Batch | LR | Angle? | Last2? | Hidden | Pooling |
|---|---|---|---|---|---|---|---|---|---|
| Vanilla | - | - | 2 | 64 | 0.0001 | - | - | 4 | mean |
| Fixed $\theta$ | ✗ | ✗ | 2 | 8 | 0.0001 | ✗ | ✓ | 4 | mean |
| Fixed $\theta$ | ✗ | ✓ | 4 | 32 | 0.0001 | ✗ | ✓ | 4 | polar |
| Fixed $\theta$ | ✓ | ✗ | 2 | 32 | 0.0001 | ✗ | ✓ | 4 | so2_mean |
| Fixed $\theta$ | ✓ | ✓ | 4 | 16 | 0.0001 | ✗ | ✓ | 2 | lie_avg |
| Learned $\theta$ | ✗ | ✗ | 2 | 16 | 0.0001 | ✓ | ✓ | 4 | mean |
| Learned $\theta$ | ✗ | ✓ | 2 | 32 | 0.0001 | ✓ | ✓ | 4 | dft |
| Learned $\theta$ | ✓ | ✗ | 2 | 8 | 0.0001 | ✓ | ✓ | 4 | polar |
| Learned $\theta$ | ✓ | ✓ | 2 | 32 | 0.0001 | ✓ | ✓ | 4 | polar |
| Fuzzy $\theta$ | ✗ | ✗ | 4 | 32 | 0.0001 | ✓ | ✓ | 2 | polar |
| Fuzzy $\theta$ | ✗ | ✓ | 2 | 16 | 0.0001 | ✗ | ✓ | 4 | dft |
| Fuzzy $\theta$ | ✓ | ✗ | 2 | 16 | 0.0001 | ✓ | ✗ | 2 | polar |
| Fuzzy $\theta$ | ✓ | ✓ | 4 | 8 | 0.0001 | ✓ | ✓ | 4 | dft |

### A.13.2 Property Retention

Table 18 provides mean squared error (MSE) of predicted properties for generated suffixes at temperature 1.0.

Table 12: Downstream hyperparameter configurations for `Tc-Ribo.`.

| Model | Stiefel | Equiv. | #Layers | Batch | LR | Angle? | Last2? | Hidden | Pooling |
|---|---|---|---|---|---|---|---|---|---|
| Vanilla | - | - | 4 | 8 | 1e-05 | - | - | 2 | mean |
| Fixed $\theta$ | ✗ | ✗ | 4 | 8 | 1e-05 | ✗ | ✗ | 2 | mean |
| Fixed $\theta$ | ✗ | ✓ | 2 | 16 | 0.0001 | ✗ | ✓ | 4 | polar |
| Fixed $\theta$ | ✓ | ✗ | 2 | 16 | 1e-05 | ✗ | ✗ | 4 | polar |
| Fixed $\theta$ | ✓ | ✓ | 4 | 16 | 1e-05 | ✗ | ✓ | 4 | mean |
| Learned $\theta$ | ✗ | ✗ | 2 | 16 | 0.0001 | ✗ | ✓ | 2 | dft |
| Learned $\theta$ | ✗ | ✓ | 2 | 16 | 0.0001 | ✓ | ✗ | 2 | polar |
| Learned $\theta$ | ✓ | ✗ | 4 | 16 | 1e-05 | ✗ | ✗ | 4 | polar |
| Learned $\theta$ | ✓ | ✓ | 4 | 8 | 1e-05 | ✓ | ✗ | 2 | polar |
| Fuzzy $\theta$ | ✗ | ✗ | 4 | 8 | 1e-05 | ✗ | ✗ | 4 | mean |
| Fuzzy $\theta$ | ✗ | ✓ | 4 | 16 | 1e-05 | ✗ | ✓ | 2 | mean |
| Fuzzy $\theta$ | ✓ | ✗ | 4 | 16 | 0.0001 | ✓ | ✓ | 4 | polar |
| Fuzzy $\theta$ | ✓ | ✓ | 4 | 16 | 0.0001 | ✓ | ✓ | 4 | mean |

Table 13: Training times for different SO(2)-based models.

| Model | Stiefel | Equiv. | Training Time |
|---|---|---|---|
| Vanilla | – | – | 1 h 51 m |
| Fixed $\theta$ | ✗ | ✗ | 2 h 23 m |
| | ✗ | ✓ | 4 h 06 m |
| | ✓ | ✗ | 7 h 35 m |
| | ✓ | ✓ | 9 h 45 m |
| Learned $\theta$ (cyclic) | ✗ | ✗ | 4 h 15 m |
| | ✗ | ✓ | 6 h 55 m |
| | ✓ | ✗ | 9 h 22 m |
| | ✓ | ✓ | 11 h 31 m |
| Fuzzy $\theta$ | ✗ | ✗ | 4 h 18 m |
| | ✗ | ✓ | 7 h 12 m |
| | ✓ | ✗ | 9 h 27 m |
| | ✓ | ✓ | 11 h 41 m |
| *25M-sequence corpus* | | | |
| EQUI-MRNA (5M) (Mamba-Transformer Hybrid) | ✓ | ✓ | 1 d 1 h 31 m |
| EQUI-MRNA (15M) (GPT-2) | ✓ | ✓ | 3 d 18 h 24 m |

Table 14: Comparison of vanilla, fixed, and learned SO(2)-based models across downstream datasets, evaluated using **Spearman correlation**. ✓indicates the presence of a component (Stiefel basis, or equivariant loss) in a configuration.

| Model | Stiefel | Equiv. | E. coli(A) | MLOS(S) | Tc-Ribo.(S) | mRFP(S) | COV Deg(S) |
|---|---|---|---|---|---|---|---|
| Vanilla | - | - | 0.580 | $0.633 \pm 0.14$ | 0.698 | 0.797 | 0.779 |
| Fixed $\theta$ | ✗ | ✗ | 0.615 | $0.602 \pm 0.11$ | 0.724 | 0.834 | 0.783 |
| | ✗ | ✓ | 0.600 | $0.633 \pm 0.16$ | 0.711 | 0.841 | 0.787 |
| | ✓ | ✗ | 0.602 | $0.667 \pm 0.19$ | 0.688 | 0.840 | 0.803 |
| | ✓ | ✓ | 0.592 | $0.571 \pm 0.086$ | 0.722 | 0.838 | 0.794 |
| Learned $\theta$ | ✗ | ✗ | 0.602 | $0.648 \pm 0.16$ | 0.672 | 0.833 | 0.799 |
| | ✗ | ✓ | 0.589 | $\mathbf{0.698} \pm \mathbf{0.15}$ | 0.698 | 0.822 | 0.787 |
| | ✓ | ✗ | **0.633** | $0.657 \pm 0.10$ | 0.701 | **0.871** | 0.790 |
| | ✓ | ✓ | 0.612 | $0.667 \pm 0.137$ | **0.741** | *0.850* | 0.797 |
| Fuzzy $\theta$ | ✗ | ✗ | 0.603 | $0.666 \pm 0.17$ | 0.680 | 0.833 | 0.786 |
| | ✗ | ✓ | 0.590 | $0.602 \pm 0.098$ | 0.682 | 0.837 | *0.805* |
| | ✓ | ✗ | *0.619* | $0.669 \pm 0.10$ | 0.729 | 0.839 | 0.793 |
| | ✓ | ✓ | 0.605 | $\mathit{0.691 \pm 0.14}$ | *0.736* | 0.844 | **0.820** |

Table 15: Comparison of vanilla, fixed, and learned SO(2)-based models across downstream datasets, evaluated using **Pearson correlation**. ✓indicates the presence of a component (Stiefel basis, or equivariant loss) in a configuration.

| Model | Stiefel | Equiv. | E. coli(AUC) | MLOS(P) | Tc-Ribo.(P) | mRFP(P) | COV Deg(P) |
|---|---|---|---|---|---|---|---|
| Vanilla | - | - | 0.4335 | $0.6203 \pm 0.157$ | 0.6451 | 0.6875 | 0.8081 |
| Fixed $\theta$ | ✗ | ✗ | 0.4290 | $0.6438 \pm 0.164$ | 0.6503 | 0.7231 | 0.8249 |
| | ✗ | ✓ | 0.4768 | $0.6508 \pm 0.140$ | 0.6245 | 0.7162 | 0.8298 |
| | ✓ | ✗ | 0.5934 | $0.7270 \pm 0.122$ | 0.6670 | 0.6979 | 0.8276 |
| | ✓ | ✓ | 0.6032 | $0.6579 \pm 0.089$ | 0.6216 | 0.6959 | 0.7939 |
| Learned $\theta$ | ✗ | ✗ | 0.4600 | $0.6412 \pm 0.139$ | 0.6086 | 0.7291 | 0.8252 |
| | ✗ | ✓ | 0.4550 | $0.6336 \pm 0.139$ | 0.6183 | 0.6911 | 0.8110 |
| | ✓ | ✗ | 0.4611 | $0.6212 \pm 0.143$ | 0.6463 | 0.7121 | 0.8166 |
| | ✓ | ✓ | 0.4749 | $0.6659 \pm 0.175$ | 0.6294 | 0.7071 | 0.8126 |
| Fuzzy $\theta$ | ✗ | ✗ | 0.4549 | $0.6684 \pm 0.183$ | 0.6036 | 0.7307 | 0.8181 |
| | ✗ | ✓ | 0.6469 | $0.6984 \pm 0.049$ | 0.5910 | 0.7027 | 0.8281 |
| | ✓ | ✗ | 0.5052 | $0.6704 \pm 0.147$ | 0.6194 | 0.7306 | 0.8087 |
| | ✓ | ✓ | 0.5452 | $0.6983 \pm 0.145$ | 0.6107 | 0.6975 | 0.8265 |

Table 16: Comparison of vanilla, fixed, and learned SO(2)-based models across downstream datasets, evaluated using **coefficient of determination** ($R^2$). ✓indicates the presence of a component (Stiefel basis, or equivariant loss) in a configuration.

| **Model** | Stiefel | Equiv. | **E. coli(F1)** | **MLOS**($R^2$) | **Tc-Ribo.**($R^2$) | **mRFP**($R^2$) | **COV Deg**($R^2$) |
|---|---|---|---|---|---|---|---|
| Vanilla | - | - | 0.4763 | $0.4037 \pm 0.196$ | 0.2240 | 0.8028 | 0.6136 |
| Fixed $\theta$ | ✗ | ✗ | 0.5340 | $0.3339 \pm 0.136$ | 0.2398 | 0.8314 | 0.6190 |
| | ✗ | ✓ | 0.5158 | $0.3345 \pm 0.133$ | 0.2467 | 0.8232 | 0.6413 |
| | ✓ | ✗ | 0.5273 | $0.3339 \pm 0.139$ | 0.2506 | 0.8140 | 0.6349 |
| | ✓ | ✓ | 0.5133 | $0.3193 \pm 0.101$ | 0.2616 | 0.7977 | 0.6007 |
| Learned $\theta$ | ✗ | ✗ | 0.5211 | $0.3573 \pm 0.153$ | 0.2446 | 0.7788 | 0.6282 |
| | ✗ | ✓ | 0.5036 | $0.2945 \pm 0.121$ | 0.2549 | 0.8038 | 0.6160 |
| | ✓ | ✗ | 0.5024 | $0.3413 \pm 0.134$ | 0.2551 | 0.8034 | 0.6215 |
| | ✓ | ✓ | 0.5153 | $0.3489 \pm 0.122$ | 0.2438 | 0.7931 | 0.5941 |
| Fuzzy $\theta$ | ✗ | ✗ | 0.5265 | $0.3715 \pm 0.162$ | 0.2373 | 0.7908 | 0.6401 |
| | ✗ | ✓ | 0.4933 | $0.3062 \pm 0.134$ | 0.2142 | 0.7851 | 0.6569 |
| | ✓ | ✗ | 0.5101 | $0.3591 \pm 0.159$ | 0.2451 | 0.7572 | 0.6210 |
| | ✓ | ✓ | 0.4940 | $0.3320 \pm 0.157$ | 0.2448 | 0.7799 | 0.6210 |

Table 17: Generation metrics and diversity (IntDiv_Gen) for SO(2)-based variants at different sampling temperatures.

| Model | Temp | Stiefel | Equiv. | FBD | Prec. | Rec. | F1 | IntDiv_Gen |
|---|---|---|---|---|---|---|---|---|
| Vanilla | 0.2 | – | – | 2561.81 | 0.093 | 0.108 | 0.0999 | 929.13 |
| Fixed $\theta$ | 0.2 | ✗ | ✗ | 2563.64 | 0.093 | 0.109 | 0.1004 | 929.11 |
| | 0.2 | ✗ | ✓ | 2565.27 | 0.093 | 0.108 | 0.0999 | 929.08 |
| | 0.2 | ✓ | ✗ | 2562.60 | 0.093 | 0.104 | 0.0982 | 929.10 |
| | 0.2 | ✓ | ✓ | 2563.44 | 0.094 | 0.107 | 0.1001 | 929.10 |
| Learned $\theta$ | 0.2 | ✗ | ✗ | 2565.05 | 0.093 | 0.107 | 0.0995 | 929.10 |
| | 0.2 | ✗ | ✓ | 2563.29 | 0.094 | 0.107 | 0.1001 | 929.12 |
| | 0.2 | ✓ | ✗ | 2565.23 | 0.093 | 0.109 | 0.1004 | 929.11 |
| | 0.2 | ✓ | ✓ | 2563.05 | 0.093 | 0.108 | 0.0999 | 929.10 |
| Fuzzy $\theta$ | 0.2 | ✗ | ✗ | 2564.17 | 0.093 | 0.104 | 0.0982 | 929.11 |
| | 0.2 | ✗ | ✓ | 2564.33 | 0.093 | 0.104 | 0.0982 | 929.11 |
| | 0.2 | ✓ | ✗ | 2562.88 | 0.093 | 0.107 | 0.0995 | 929.10 |
| | 0.2 | ✓ | ✓ | 2563.90 | 0.093 | 0.103 | 0.0977 | 929.12 |
| Equi-mRNA (5M) | 0.2 | ✓ | ✓ | 819.18 | 0.680 | 0.745 | 0.7110 | 1045.95 |
| Equi-mRNA (25M) | 0.2 | ✓ | ✓ | 802.89 | 0.519 | 0.519 | 0.5190 | 965.27 |
| Vanilla | 0.4 | – | – | 2561.99 | 0.093 | 0.108 | 0.0999 | 929.15 |
| Fixed $\theta$ | 0.4 | ✗ | ✗ | 2564.09 | 0.094 | 0.107 | 0.1001 | 929.12 |
| | 0.4 | ✗ | ✓ | 2564.21 | 0.093 | 0.109 | 0.1004 | 929.11 |
| | 0.4 | ✓ | ✗ | 2562.49 | 0.093 | 0.109 | 0.1004 | 929.12 |
| | 0.4 | ✓ | ✓ | 2562.72 | 0.093 | 0.108 | 0.0999 | 929.12 |
| Learned $\theta$ | 0.4 | ✗ | ✗ | 2563.73 | 0.093 | 0.108 | 0.0999 | 929.13 |
| | 0.4 | ✗ | ✓ | 2562.55 | 0.094 | 0.108 | 0.1005 | 929.14 |
| | 0.4 | ✓ | ✗ | 2564.46 | 0.093 | 0.109 | 0.1004 | 929.15 |
| | 0.4 | ✓ | ✓ | 2562.82 | 0.093 | 0.108 | 0.0999 | 929.12 |
| Fuzzy $\theta$ | 0.4 | ✗ | ✗ | 2564.21 | 0.093 | 0.108 | 0.0999 | 929.12 |
| | 0.4 | ✗ | ✓ | 2563.56 | 0.093 | 0.106 | 0.0991 | 929.13 |
| | 0.4 | ✓ | ✗ | 2562.74 | 0.093 | 0.108 | 0.0999 | 929.12 |
| | 0.4 | ✓ | ✓ | 2562.79 | 0.093 | 0.110 | 0.1008 | 929.13 |
| Equi-mRNA (5M) | 0.4 | ✓ | ✓ | 579.21 | 0.726 | 0.777 | 0.7506 | 1049.83 |
| Equi-mRNA (25M) | 0.4 | ✓ | ✓ | 310.40 | 0.883 | 0.877 | 0.8800 | 984.58 |
| Vanilla | 0.6 | – | – | 2562.24 | 0.093 | 0.110 | 0.1008 | 929.13 |
| Fixed $\theta$ | 0.6 | ✗ | ✗ | 2564.36 | 0.094 | 0.107 | 0.1001 | 929.13 |
| | 0.6 | ✗ | ✓ | 2563.80 | 0.093 | 0.110 | 0.1008 | 929.13 |
| | 0.6 | ✓ | ✗ | 2562.11 | 0.093 | 0.110 | 0.1008 | 929.13 |
| | 0.6 | ✓ | ✓ | 2062.68 | 0.120 | 0.134 | 0.1266 | 936.00 |
| Learned $\theta$ | 0.6 | ✗ | ✗ | 2563.25 | 0.093 | 0.108 | 0.0999 | 929.15 |
| | 0.6 | ✗ | ✓ | 2562.04 | 0.094 | 0.108 | 0.1005 | 929.15 |
| | 0.6 | ✓ | ✗ | 2564.53 | 0.093 | 0.109 | 0.1004 | 929.15 |
| | 0.6 | ✓ | ✓ | 2562.71 | 0.093 | 0.108 | 0.0999 | 929.13 |
| Fuzzy $\theta$ | 0.6 | ✗ | ✗ | 2563.20 | 0.093 | 0.108 | 0.0999 | 929.13 |
| | 0.6 | ✗ | ✓ | 2462.77 | 0.099 | 0.116 | 0.1068 | 930.43 |
| | 0.6 | ✓ | ✗ | 2563.15 | 0.093 | 0.108 | 0.0999 | 929.12 |
| | 0.6 | ✓ | ✓ | 2562.95 | 0.093 | 0.111 | 0.1012 | 929.14 |
| Equi-mRNA (5M) | 0.6 | ✓ | ✓ | 341.68 | 0.791 | 0.817 | 0.8038 | 1049.53 |
| Equi-mRNA (25M) | 0.6 | ✓ | ✓ | 196.15 | 0.959 | 0.954 | 0.9565 | 1001.13 |
| Vanilla | 0.8 | – | – | 2562.67 | 0.093 | 0.112 | 0.1016 | 929.15 |
| Fixed $\theta$ | 0.8 | ✗ | ✗ | 2564.74 | 0.094 | 0.107 | 0.1001 | 929.14 |
| | 0.8 | ✗ | ✓ | 1714.65 | 0.142 | 0.145 | 0.1435 | 943.15 |
| | 0.8 | ✓ | ✗ | 2562.17 | 0.093 | 0.110 | 0.1008 | 929.15 |
| | 0.8 | ✓ | ✓ | 1039.52 | 0.199 | 0.201 | 0.2000 | 954.47 |
| Learned $\theta$ | 0.8 | ✗ | ✗ | 2564.03 | 0.093 | 0.107 | 0.0995 | 929.15 |
| | 0.8 | ✗ | ✓ | 1525.08 | 0.147 | 0.152 | 0.1495 | 945.01 |
| | 0.8 | ✓ | ✗ | 2564.30 | 0.093 | 0.109 | 0.1004 | 929.15 |
| | 0.8 | ✓ | ✓ | 2562.60 | 0.093 | 0.108 | 0.0999 | 929.14 |
| Fuzzy $\theta$ | 0.8 | ✗ | ✗ | 2374.21 | 0.097 | 0.122 | 0.1081 | 930.52 |
| | 0.8 | ✗ | ✓ | 1164.22 | 0.165 | 0.166 | 0.1655 | 949.01 |
| | 0.8 | ✓ | ✗ | 2562.80 | 0.093 | 0.108 | 0.0999 | 929.12 |
| | 0.8 | ✓ | ✓ | 2562.93 | 0.093 | 0.111 | 0.1012 | 929.15 |
| Equi-mRNA (5M) | 0.8 | ✓ | ✓ | 129.43 | 0.895 | 0.909 | 0.9019 | 1050.24 |
| Equi-mRNA (25M) | 0.8 | ✓ | ✓ | 172.02 | 0.977 | 0.973 | 0.9750 | 1009.09 |
| Vanilla | 1.0 | – | – | 2562.78 | 0.093 | 0.112 | 0.1016 | 929.16 |
| Fixed $\theta$ | 1.0 | ✗ | ✗ | 2564.44 | 0.094 | 0.111 | 0.1018 | 929.14 |
| | 1.0 | ✗ | ✓ | 964.80 | 0.209 | 0.215 | 0.2120 | 959.16 |
| | 1.0 | ✓ | ✗ | 2562.25 | 0.093 | 0.110 | 0.1008 | 929.15 |
| | 1.0 | ✓ | ✓ | 653.14 | 0.350 | 0.355 | 0.3525 | 969.65 |
| Learned $\theta$ | 1.0 | ✗ | ✗ | 1516.35 | 0.138 | 0.144 | 0.1409 | 941.34 |
| | 1.0 | ✗ | ✓ | 958.41 | 0.204 | 0.212 | 0.2079 | 958.78 |
| | 1.0 | ✓ | ✗ | 1921.42 | 0.132 | 0.143 | 0.1373 | 940.06 |
| | 1.0 | ✓ | ✓ | 1807.64 | 0.123 | 0.141 | 0.1314 | 939.07 |
| Fuzzy $\theta$ | 1.0 | ✗ | ✗ | 1520.08 | 0.127 | 0.150 | 0.1375 | 940.55 |
| | 1.0 | ✗ | ✓ | 580.19 | 0.369 | 0.367 | 0.3680 | 969.57 |
| | 1.0 | ✓ | ✗ | 2023.60 | 0.106 | 0.121 | 0.1130 | 933.61 |
| | 1.0 | ✓ | ✓ | 1652.77 | 0.136 | 0.156 | 0.1453 | 941.85 |
| Equi-mRNA (5M) | 1.0 | ✓ | ✓ | 76.13 | 0.977 | 0.976 | 0.9765 | 1051.16 |
| Equi-mRNA (25M) | 1.0 | ✓ | ✓ | 177.77 | 0.987 | 0.984 | 0.9855 | 1014.56 |

Table 18: MSE of predicted property across model variants and datasets at sampling temperature 1.0.

| Model | Dataset | Stiefel | Equiv. | MSE Pred. |
|---|---|---|---|---|
| Vanilla | icodon | – | – | 0.4640 |
| | mlos0 | – | – | 0.1894 |
| | mrfp | – | – | 1.8485 |
| | switch | – | – | 0.4141 |
| | deg | – | – | 0.7868 |
| Fixed $\theta$ | icodon | ✗ | ✗ | 0.4649 |
| | mlos0 | ✗ | ✗ | 0.1888 |
| | mrfp | ✗ | ✗ | 1.8485 |
| | switch | ✗ | ✗ | 0.4153 |
| | deg | ✗ | ✗ | 0.7921 |
| Fixed $\theta$ | icodon | ✗ | ✓ | 0.4346 |
| | mlos0 | ✗ | ✓ | 0.1597 |
| | mrfp | ✗ | ✓ | 1.7898 |
| | switch | ✗ | ✓ | 0.4278 |
| | deg | ✗ | ✓ | 0.6716 |
| Fixed $\theta$ | icodon | ✓ | ✗ | 0.4640 |
| | mlos0 | ✓ | ✗ | 0.1888 |
| | mrfp | ✓ | ✗ | 1.8478 |
| | switch | ✓ | ✗ | 0.4069 |
| | deg | ✓ | ✗ | 0.8017 |
| Fixed $\theta$ | icodon | ✓ | ✓ | 0.4252 |
| | mlos0 | ✓ | ✓ | 0.1426 |
| | mrfp | ✓ | ✓ | 1.7786 |
| | switch | ✓ | ✓ | 0.3749 |
| | deg | ✓ | ✓ | 0.6215 |
| Learned $\theta$ | icodon | ✗ | ✗ | 0.4470 |
| | mlos0 | ✗ | ✗ | 0.1684 |
| | mrfp | ✗ | ✗ | 1.8193 |
| | switch | ✗ | ✗ | 0.4221 |
| | deg | ✗ | ✗ | 0.7086 |
| Learned $\theta$ | icodon | ✗ | ✓ | 0.4325 |
| | mlos0 | ✗ | ✓ | 0.1695 |
| | mrfp | ✗ | ✓ | 1.7806 |
| | switch | ✗ | ✓ | 0.3623 |
| | deg | ✗ | ✓ | 0.6733 |
| Learned $\theta$ | icodon | ✓ | ✗ | 0.4509 |
| | mlos0 | ✓ | ✗ | 0.1791 |
| | mrfp | ✓ | ✗ | 1.8256 |
| | switch | ✓ | ✗ | 0.4155 |
| | deg | ✓ | ✗ | 0.7357 |
| Learned $\theta$ | icodon | ✓ | ✓ | 0.4498 |
| | mlos0 | ✓ | ✓ | 0.1776 |
| | mrfp | ✓ | ✓ | 1.8216 |
| | switch | ✓ | ✓ | 0.4091 |
| | deg | ✓ | ✓ | 0.7523 |
| Fuzzy $\theta$ | icodon | ✗ | ✗ | 0.4474 |
| | mlos0 | ✗ | ✗ | 0.1642 |
| | mrfp | ✗ | ✗ | 1.8140 |
| | switch | ✗ | ✗ | 0.3992 |
| | deg | ✗ | ✗ | 0.7291 |
| Fuzzy $\theta$ | icodon | ✗ | ✓ | 0.4296 |
| | mlos0 | ✗ | ✓ | 0.1361 |
| | mrfp | ✗ | ✓ | 1.7620 |
| | switch | ✗ | ✓ | 0.3872 |
| | deg | ✗ | ✓ | 0.6319 |
| Fuzzy $\theta$ | icodon | ✓ | ✗ | 0.4582 |
| | mlos0 | ✓ | ✗ | 0.1786 |
| | mrfp | ✓ | ✗ | 1.8332 |
| | switch | ✓ | ✗ | 0.4122 |
| | deg | ✓ | ✗ | 0.7707 |
| Fuzzy $\theta$ | icodon | ✓ | ✓ | 0.4497 |
| | mlos0 | ✓ | ✓ | 0.1758 |
| | mrfp | ✓ | ✓ | 1.8179 |
| | switch | ✓ | ✓ | 0.4171 |
| | deg | ✓ | ✓ | 0.7149 |

