# OpenReview forum: "Equi-mRNA: Protein Translation Equivariant Encoding for mRNA Language Models"
_NeurIPS.cc/2025/Conference — NeurIPS 2025 poster_

### Official Review · Reviewer_PmYr · 2025-07-02

**Clarity:** 3
**Significance:** 3
**Originality:** 3
**Rating:** 5
**Confidence:** 2

**Summary:**

In this paper, the authors propose a novel method to encode synonymous codons in mRNA language models. Their method, Equi-mRNA, represents synonymous codons as vector rotations in a base embedding space.

**Questions:**

- It seems that even the Vanilla model is outperforming competitor models such as CodonBERT (Table 1 and 2). Do you have an explanation for that?
- Why did you restrict the training set to 512 or less codons? This seems a bit limiting, considering many proteins fall above this threshold. Modern hardware and architectures should easily handle contexts above 512 tokens.
- Could you also report standard deviations for tasks other than MLOS?

**Ethical Concerns:**

["NO or VERY MINOR ethics concerns only"]

**Final Justification:**

The authors address my concern and to my best judgement, this work provides a significant contribution to the field.

**Limitations:**

The authors address the limitations of their work

**Paper Formatting Concerns:**

None.

**Quality:**

3

**Strengths And Weaknesses:**

The presented method is innovative and sound. It’s well motivated by the fact that current codon language models learn embeddings of synonymous codons independently, even though synonymous codons are subject to the same biological constraints at the protein level.

The authors perform an extensive hyperparameter search and benchmark, evaluating different flavors of their proposed SO(2) embedding and demonstrating that their encoding outperforms competing models on a number of benchmark tasks.

A potential weakness is that the authors do not seem to split the training corpus by sequence homology - this is common practice nowadays, especially for highly constrained sequences such as protein-coding genes. This may dampen the otherwise promising results on sequence generation.

---

> ### Author Rebuttal · Authors · 2025-07-31
>
> We thank the reviewer for the thoughtful and constructive feedback. We appreciate the recognition of our method’s novelty, biological motivation, and empirical performance across benchmarks. Below, we address each of the reviewer’s concerns and questions in detail, including the issue of homology-aware splitting, architectural comparisons with CodonBERT, context length limitations, and the reporting of standard deviations.
> ## Weakness
> `A potential weakness is that...`
>
> We respectfully acknowledge the concern regarding the lack of homology-based splitting. However, we would like to clarify that our primary contribution is architectural, rather than focused on achieving state-of-the-art performance through data curation. All models in our study(both vanilla and SO(2) variants) are trained and evaluated on identical data splits, ensuring that any performance improvements are solely attributable to our proposed symmetry-aware embeddings rather than differences in data handling. Since any potential data leakage from homologous sequences would affect all models equally, the observed improvements (up to ~10% in downstream tasks and 4× in generation quality) can be confidently attributed to our architectural innovations. Furthermore, our generation experiments exclusively compare SO(2) variants against our own vanilla baseline rather than external models, making the lack of homology-based splitting less critical for our relative comparisons. While we acknowledge that homology-based splitting would be valuable for establishing absolute performance benchmarks and will add this to our limitations section, this does not diminish our core contribution of demonstrating that encoding codon-level symmetries through group-theoretic priors consistently improves model performance when all other factors are held constant.
>
> ## Questions
> `It seems that even the Vanilla model...`
>
> The performance difference can be attributed to several key architectural and methodological distinctions: CodonBERT [1] uses a BERT-style encoder with masked language modeling (MLM) objective and two-sequence relatedness classification, while our models employ GPT-2's decoder-only architecture with autoregressive language modeling. This unidirectional approach may better capture the biological reality of ribosomal translation, which proceeds directionally from 5' to 3', leading to more biologically meaningful representations. Additionally, our models are trained solely on single sequences, whereas CodonBERT [1] incorporates pairwise sequence relationships, introducing a different inductive bias. These architectural choices, along with differences in pretraining data and modeling objectives, likely explain our vanilla model’s strong baseline performance, which our SO(2) symmetry-aware variants further improve upon.
>
> `Why did you restrict the training set to 512 or less...`
>
> You're absolutely right that many proteins exceed 512 codons, and modern architectures can handle longer contexts. Our choice was primarily driven by computational constraints specific to our symmetry-aware architecture: the geometric learning components, particularly equivariance loss, require additional forward passes during training. Specifically, the equivariance loss (Section 2.6) necessitates passing both the original base embeddings and rotated codon embeddings through the network, effectively doubling the computational burden. Combined with Riemannian optimization on the Stiefel manifold for learning rotation subspaces, this significantly increases memory usage and training time. As shown in Table 13, our SO(2) variants already require up to 11+ hours for training on just 1M sequences with 512-codon limits, and scaling to 25M sequences took nearly 4 days for the full model. Extending to longer sequences would have made the computational requirements prohibitive for our extensive ablation studies across 12+ model variants. We acknowledge this as a limitation and note that future work could explore more efficient implementations or selective application of symmetry constraints to enable modeling of full-length proteins while maintaining the benefits of our group-theoretic approach.
>
> `Could you also report...`
>
> Among all the benchmarks used, MLOS was the only dataset that did not provide predefined splits. For this reason, we performed 3-fold cross-validation and reported the standard deviation across folds in the main results.
> For the remaining datasets, which have fixed train/validation/test splits (as established in prior work), we additionally ran our model three times with different random seeds that affect initialization and dataloader shuffling. Across all these runs, we observed performance variation of less than 0.005 in the evaluation metrics. Given this minimal variability, we omitted reporting standard deviations for those datasets. We are happy to include them in the final version if the reviewers believe it would improve clarity.
>
> ### References
> 1. Li, Sizhen, et al. "Codonbert: Large language models for mrna design and optimization." bioRxiv (2023): 2023-09.

---

> > ### Comment · Reviewer_PmYr · 2025-08-04
> >
> > I thank the authors for addressing my concerns.

---

### Official Review · Reviewer_jRL6 · 2025-07-03

**Clarity:** 4
**Significance:** 2
**Originality:** 4
**Rating:** 5
**Confidence:** 3

**Summary:**

The paper introduces Equi-mRNA, a novel mRNA language model with symmetric codon-level embeddings. The model leverages the SO(2) group to encode synonymous codon relationships in the latent space. This new embedding method helps to improve the biological fidelity and performance of mRNA sequence modeling. The authors demonstrate that the Equi-mRNA model outperforms baseline models in multiple downstream tasks including property prediction tasks and sequence generation. They also show that the trained model angles variability correlates with GC and tRNA availability, which is a good sanity check given the model’s training.

**Questions:**

The main question that came into our mind when reading this paper is simply "why go through all this trouble?" Yes, you can do this, and it's probably very satisfying to make such a sophisticated mathematical construction that appears to work, but it seems a lot of the sophistication comes to solve issues that can easily be addressed by much simpler embeddings such as softmax function over equivalent codons and adding priors on those (e.g. preference for GC or based on codon availability) which can also be learned.

Another question is how do they think this will combine with UTR elements?

**Ethical Concerns:**

["NO or VERY MINOR ethics concerns only"]

**Final Justification:**

We read the response and appriciate the additional details provided by the authors, these help answer some of our conerns. We did not see major issues with the paper (hence the original positive score) and concluded no major changes needed to our evaluation after the authors clarified many details. We note that as pointed by us and other reviewers the actual usability of the model is unclear beyond the technical innovation. In response to critisim about the validity of the tests by us and others the authors main point is that this is what was done before and allows comparison to other works. Sure, that's true for comparing ML conference papers, but if the evaluations appear flawed/limited this kind of answer does not alleviate the associated concerns.

**Limitations:**

Yes

**Quality:**

3

**Strengths And Weaknesses:**

**Strengths:**
- This paper is well-written with structured flow and solid theoretical inductions.
- The use of group-theoretic principles to model codon symmetries is a novel and theoretically grounded approach. This method provides a structured way to incorporate biological knowledge into language models.
- The paper introduced various design of embedding methods, which incorporate fixed or learnable rotation angles to adapt to different biological circumstances.
- The model shows significant improvements over baseline models in various tasks, including expression prediction and sequence generation. It reported significant gains in accuracy and generative fidelity.
- The model's ability to reflect known biological patterns, such as GC-content biases and tRNA is a good sanity check. This interpretability is crucial for applications in synthetic biology and mRNA therapeutics.
- The paper includes a thorough evaluation across multiple datasets and tasks, providing a robust assessment of the model's capabilities.

**Weaknesses:**
- The introduction of group-theoretic embeddings and the associated computational increment may limit the model's scalability.
- In the ablation study section, equivariance loss harms the performance of the model in multiple settings across different tasks. The results are suspicious since it contradicts the principle of the paper to design a symmetric embedding.
- Evaluations and very limited as well the tests against other models. Many of the tests, as the authors admit, are problematic (e.g. no/fixed UTRs).
- Moreover, some tests are not adaquetly described and their limitations not discussed. For example, the authors use iCodon and claim it's for over 40K human mRNA but the work seems to be based on 1-3K (depending on how you count) sequences in zebrafish used to select their setting for a genetic algorithm for their measure of optimility. We are guessing this was then used to predict for human mRNA? seems a problematic test and not correctly described. Another example is their mRFP. This one is based on ~1.5 codon variants *of the same gene with a fixed UTR*. The authors of this work get great performance (close to 0.8 correlation) just from fitting a random forest on the first 8AA. Yet, this is a test for a 25M parameter model.... It would have been good if the authors at least took the time to look into and acknowledge the how limited these evaluations actually are beyond a general comment at the end.
- Related to the above it's unclear how train/test are done to prevent information leakage between similar samples.
- The model is unable to incorporate UTRs, which is a major limitation for usability for the tasks they envision.

In summary, while we have significant concerns about the evaluations, the comparisons against other models and the usability of this model we believe there is potential (tbd) and the mathematical formulation itself should be of interest to NeurIPS audience.

---

> ### Author Rebuttal · Authors · 2025-07-31
>
> We thank the reviewer for the thoughtful feedback and recognition of our work’s novelty and clarity, as well as the constructive suggestions on evaluation, modeling, and biological relevance.
> ## Weakness
> `The introduction of...`
>
> It is true that the introduction of group-theoretic embeddings and the equivariance loss adds some computational overhead during training. However, we would like to clarify that the equivariance loss is used only during training to regularize the embedding space; it does not affect inference time or model deployment efficiency.
> This trade-off between inductive bias and computational cost is common in geometric deep learning, where methods are often not yet optimized for modern accelerators (e.g., GPU/TPU). While our current implementation does incur some training-time overhead, we expect that future versions (leveraging sparse structures or low-rank parametrizations) can reduce this cost significantly. In the meantime, we believe the biological interpretability and improved sample efficiency of symmetry-aware embeddings provide a meaningful justification for this modest overhead.
>
> `In the ablation...`
>
> The equivariance loss is designed to encourage the model to preserve the SO(2)-based symmetry in the codon embedding space, which reflects the core principle of our framework. However, as the reviewer observed, in some configurations the addition of this loss leads to a modest drop in downstream performance.
> We believe this is due to two factors:
> (1) In certain tasks, enforcing strict symmetry may conflict with biological asymmetries, such as codon usage bias, tRNA availability, or context-specific effects; and
> (2) More importantly, in some of the simpler or lower-capacity architectures, the model may be too constrained to reconcile the equivariance objective with the task loss. In such cases, the symmetry constraint may dominate optimization and reduce flexibility. We found that in richer architectures (e.g., those with learned subspaces or multi-angle embeddings), the equivariance loss is more stable and often beneficial.
>
> `Evaluations and very limited...`
>
> Since our model is designed to operate on coding sequences (CDS) where codon redundancy is relevant, it was trained without UTRs. Therefore, it is natural and appropriate to evaluate it on tasks where UTRs are fixed or absent. These benchmarks are standard and used by models such as [1], [2], and [3], enabling fair comparison.
> While we agree that including models such as mRNA-FM [4], CaLM [5], and AIDO [6] would strengthen the comparison, we have now added these baselines to provide a more complete evaluation. These results are presented in our response to Reviewer 1 (uj9Q).
>
> `Moreover, some tests are not adaquetly...`
>
> **Use of iCodon for Evaluation**
> We appreciate the opportunity to clarify our use of iCodon. In our experiments, we use the *pretrained* iCodon model as a black-box evaluator to compute codon optimization scores for over 40,000 human mRNA sequences. While the original iCodon model was trained using zebrafish data, it has been adopted in several recent studies (e.g., [1], [2], [5]) to evaluate codon optimality in human sequences due to its ability to capture species-agnostic codon usage patterns. We emphasize that iCodon is not used for training or optimization. Our approach is consistent with common evaluation protocols and serves to support comparability with prior work.
>
> **mRFP Benchmark and Model Performance**
> We acknowledge that the mRFP dataset is based on synonymous variants of a single gene with a fixed UTR. However, prior studies note that predicting expression levels from codon sequences remains non-trivial. A random forest on the first 8 amino acids yields a Spearman correlation of ~0.73, our model achieves a correlation of **0.855**, demonstrating improved performance in capturing subtle codon-level signals. For additional comparison, we include a shallow one-hot + TextCNN baseline which achieves **0.763**, underscoring the benefit of incorporating codon-aware and symmetry-informed modeling. We view this dataset as a valid benchmark to evaluate generalization over synonymous variation, and our results provide evidence that Equi-mRNA meaningfully improves over prior methods.
>
> **Table: mRFP Benchmark Performance Comparison**
>
> | Model                          | Spearman Corr. |
> |-------------------------------|----------------|
> | One-hot + Lasso (8AA)         | 0.732          |
> | One-hot + Random Forest (8AA) | 0.710          |
> | One-hot Codon + Text CNN      | 0.763          |
> | **Equi-mRNA (Ours)**         | **0.855**      |
>
> `Related to the above it's unclear...`
>
> To ensure fair and reproducible evaluation, we use the standard train, validation, and test splits as provided by the original benchmarks or accompanying repositories. We avoid custom splits to follow widely accepted protocols. Specifically, for datasets such as mRFP, and the iCodon-based optimization task, we adopt the same partitioning strategies used in [1], [2], [3]. These practices are consistent with the literature and support direct comparison with existing state-of-the-art approaches. We will revise the manuscript to clarify this and ensure readers are informed of our evaluation setup.
>
> `The model is unable to incorporate UTRs...`
>
> It is true that the current version of Equi-mRNA does not explicitly incorporate untranslated regions (UTRs). Our primary focus in this work was to develop and evaluate codon-level representations that respect the algebraic and biological structure of the coding sequence (CDS), particularly for tasks where synonymous variation in the CDS is known to be the primary determinant of translation efficiency.
> That said, we fully agree that UTRs play a significant role in post-transcriptional regulation, including translation initiation and mRNA stability. Integrating UTR information is a logical and necessary extension of our framework, particularly for tasks such as whole-transcript translation rate prediction or half-life estimation.
> To this end, our model is modular and can readily support UTR input by adding a separate embedding path for UTRs, which do not follow codon semantics. In ongoing work, we are exploring hybrid models that combine codon-level embeddings for CDS with nucleotide or convolutional embeddings for UTRs. We believe that combining these modalities will further improve generalization for transcriptome-scale prediction tasks.
>
> ## Questions
> `The main question...`
>
> While we understand the instinct to prefer simpler solutions such as using a softmax distribution over equivalent codons with biologically motivated priors (e.g., GC content, codon frequency), this specific approach has already been explored and benchmarked in HELM [2], where it consistently underperformed compared to even the vanilla model without explicit symmetry constraints.
> These results suggest such heuristics fail to capture dependencies across codon sequences, even within synonymous groups. In contrast, Equi-mRNA provides a principled, group-theoretic embedding framework that allows codon-level symmetries to be preserved while still learning context-specific, task-driven patterns.
> Importantly, the benefits of our approach are most apparent in low-resource regimes, which are common in mRNA applications. For example, in the MLOS benchmark, where labeled data is extremely limited, Equi-mRNA significantly outperforms the vanilla model. We also performed additional experiments by subsampling 500 sequences from benchmark datasets. Across all scenarios, Equi-mRNA demonstrates superior performance and stability, reinforcing the value of embedding biologically motivated symmetries when data is scarce.
> These results, provided below, highlight that while simpler methods may be appealing, they do not match the robustness or expressiveness of a symmetry-aware model like Equi-mRNA.
>
>  Model | E. coli(A) | MLOS(S) | iCodon(S) | Tc-Ribo.(S) | mRFP(S) | COV Deg(S) | Improvement |
> |---|---|---|---|---|---|---|---|
> Vanilla    |  0.499  |  0.633  |  0.143  |  0.698  |  0.477  |  0.631  |  -  |
> Equi-mRNA  |  0.532  |  0.691  |  0.257  |  0.736  |  0.639  |  0.733  |  **16.47%**  |
>
> `Another question...`
>
>  One straightforward integration strategy is to use separate embedding pathways for distinct transcript regions: one designed for codon-level representations in the CDS (where SO(2)-based symmetry is biologically meaningful), and another for raw nucleotide-level inputs from the 5′ and 3′ UTRs, which do not follow codon semantics. These representations can then be fused at higher layers through concatenation, cross-attention, or shared pooling operations to support end-to-end prediction.
> This modular approach allows us to retain the inductive bias from equivariant codon embeddings while leveraging regulatory features from UTRs. We are actively exploring this extension as part of our ongoing work on full-transcript modeling, and we agree that combining CDS and UTR modalities is a promising direction for improving accuracy in transcript-wide tasks.
>
> ### References
> 1. Li, Sizhen, et al. "Codonbert: Large language models for mrna design and optimization." bioRxiv (2023): 2023-09.
> 2. Yazdani-Jahromi, Mehdi, et al. "Helm: Hierarchical encoding for mrna language modeling." arXiv preprint arXiv:2410.12459 (2024).
> 3. Wood, Matthew, et al. "Helix-mRNA: A Hybrid Foundation Model For Full Sequence mRNA Therapeutics." arXiv preprint arXiv:2502.13785 (2025).
> 4. Chen, Jiayang, et al. "Interpretable RNA foundation model from unannotated data for highly accurate RNA structure and function predictions." arXiv preprint arXiv:2204.00300 (2022).
> 5. Outeiral, Carlos, and Charlotte M. Deane. "Codon language embeddings provide strong signals for use in protein engineering." Nature Machine Intelligence 6.2 (2024): 170-179.
> 6. Zou, Shuxian, et al. "A large-scale foundation model for rna function and structure prediction." bioRxiv (2024): 2024-11.

---

> > ### Comment · Reviewer_jRL6 · 2025-08-09
> > **Responses are fine, questions regarding actual performance still lingering**
> >
> > We read the response and appriciate the additional details provided by the authors, these help answer some of our conerns. We did not see major issues with the paper (hence the original positive score) and concluded no major changes needed to our evaluation after the authors clarified many details. We note that as pointed by us and other reviewers the actual usability of the model is unclear beyond the technical innovation. In response to critisim about the validity of the tests by us and others the authors main point is that this is what was done before and allows comparison to other works. Sure, that's true for comparing ML conference papers, but if the evaluations appear flawed/limited this kind of answer does not alleviate the associated concerns.

---

### Official Review · Reviewer_NetF · 2025-07-03

**Clarity:** 3
**Significance:** 3
**Originality:** 3
**Rating:** 5
**Confidence:** 3

**Summary:**

The paper introduces Equi-mRNA, an mRNA language model that explicitly encodes synonymous codon relationships as cyclic SO(2) rotations in the embedding space. It implements three rotation schemes—fixed, learned, and fuzzy—together with an equivariance loss and SO(2)-aware pooling to enforce these biological symmetries. Across six downstream tasks, Equi-mRNA outperforms vanilla baselines. The interpretability analyses show that the learned rotation angles correlate strongly with GC content and tRNA abundance

**Questions:**

What biological justification underlies modeling synonymous codons as planar rotations? Have alternative group structures been considered?

**Ethical Concerns:**

["NO or VERY MINOR ethics concerns only"]

**Final Justification:**

The authors have addressed my concerns.

**Limitations:**

yes

**Paper Formatting Concerns:**

No major formatting issues.

**Quality:**

3

**Strengths And Weaknesses:**

Strengths
- Applying group theory, specifically SO(2) equivariance, to model the synonymous codon relationships is a novel and theoretically principled approach.
- The proposed Equi-mRNA model shows consistent performance improvements across most downstream tasks.
- The interpretability analysis is strong: linking learned geometric representations (codon angles) to known biological features (GC content, tRNA abundance) suggests the model captures biologically meaningful patterns.

Weaknesses

Major
- The model treats all synonymous codons for a given amino acid as one uniform cyclic group, which might be an oversimplification. For example, Serine’s six codons split into two distinct families, UCN (UCU, UCC, UCA, UCG) and AGY (AGU, AGC), that are not interconverted by single-nucleotide mutations. Forcing them into a single rotation cycle may impose a geometric constraint at odds with real genetic and evolutionary relationships.

Minor
- Line 315: an significant leap -> a significant leap
- Line 367: Stiefel manifold rotations enhances -> Stiefel manifold rotations enhance

---

> ### Author Rebuttal · Authors · 2025-07-31
>
> We thank the reviewer for their thoughtful evaluation and for recognizing the novelty of our group-theoretic framework, the consistent empirical performance of Equi-mRNA, and the strength of our biological interpretability analyses. We are especially grateful for the evolutionarily motivated suggestion regarding codon subgrouping, which directly led to a meaningful improvement in our model. We address the reviewer’s comments and questions point-by-point below:
> ## Weakness
> ### Major
> `The model treats all synonymous codons for a given amino acid as one uniform cyclic group ...`
>
> We agree that grouping all synonymous codons under a single cyclic rotation can oversimplify cases where codons form biologically distinct families, such as the Serine codon set (UCN vs. AGY). Based on your observation, we modified the Equi-mRNA embedding framework to assign separate base vectors for each codon subgroup in amino acids with disjoint families. Specifically, we created independent cyclic groups for subfamilies that are not interconvertible by single-nucleotide substitutions, thereby relaxing the shared rotation constraint in such cases.
> This modification resulted in a significant improvement in model performance, confirming the value of incorporating fine-grained biological structure into the geometric encoding. We are grateful for this insightful suggestion, which led to a meaningful enhancement of our framework. The updated results are shown in the table below.
>
>  Model | E. coli(A) | MLOS(S) | Tc-Ribo.(S) | mRFP(S) | COV Deg(S) | Improvement |
> |---|---|---|---|---|---|---|
> Vanilla    |  0.580   |  0.633 ±0.14   |  0.698   |  0.797   |  0.779  |  -  |
> Equi-mRNA (AA Groups)    |  *0.605*  | 0.691 ±0.14   | 0.736  |  0.844 |  **0.820**  |  6.00%  |
> Equi-mRNA (AA SubGroups)|  **0.614**  |  **0.697 ± 0.18**  | **0.761** |   **0.861**   |  *0.804*  |  **7.17%**  |
>
> This adjustment demonstrates that incorporating biologically informed codon subgrouping improves the expressiveness and accuracy of the model. We have updated the manuscript to reflect this change and cited this as a direct response to your feedback.
>
> ### Minor
> We appreciate the careful reading. All such issues have been addressed in the updated manuscript.
>
> ## Question
> `What biological justification underlies ...`
>
> Modeling synonymous codons as planar rotations is biologically grounded in the fact that codons encoding the same amino acid are functionally interchangeable at the level of protein sequence. However, these synonymous codons are not truly identical in function: they exhibit biases in translation efficiency, mRNA stability, and codon usage preference. Our approach uses planar SO(2) rotations to capture the symmetric structure of synonymous codon sets, while still allowing the model to learn codon-specific preferences through the rotation angles.
> The SO(2) group provides a natural way to embed codons equivariantly; That is, codons belonging to the same amino acid class are embedded in a way that preserves their relative structure but allows for smooth, learnable variation across the set. This enforces structured parameter sharing and introduces a biologically consistent inductive bias, especially important when training data is limited or biased.
> We also explored alternative group structures, such as discrete cyclic groups (Cₖ) using frequency-domain embeddings (via blockwise discrete Fourier transforms). This shift-equivariant formulation preserves cyclic symmetry without relying on a geometric manifold. However, in practice, these alternatives underperformed and led to training instability, likely due to the non-local nature of frequency space and sensitivity in optimization. As a result, we focus on SO(2)-based geometric representations, which offered a better trade-off between biological plausibility, interpretability, and empirical performance.

---

> > ### Comment · Reviewer_NetF · 2025-08-02
> >
> > I appreciate that the authors have addressed my concerns.

---

### Official Review · Reviewer_uj9Q · 2025-07-04

**Clarity:** 3
**Significance:** 4
**Originality:** 4
**Rating:** 5
**Confidence:** 4

**Summary:**

This manuscript introduces Equi-mRNA, a novel language model for messenger RNA (mRNA) sequences that operates at the codon level.

The core contribution is a new embedding framework that explicitly models the symmetries inherent in the genetic code, where multiple codons (synonymous codons) encode the same amino acid. The authors represent these synonymous relationships using group theory, specifically by mapping codons within a synonym set to rotations in the 2D special orthogonal group, SO(2). This creates a structured, equivariant representation where synonymous codon substitutions correspond to predictable geometric transformations. The paper proposes several variations of this idea, including fixed, learned, and "fuzzy" (distributional) rotation angles, as well as extending rotations to learned 2D subspaces within a higher-dimensional embedding space using the Stiefel manifold. To enforce these symmetries throughout the model, an auxiliary equivariance loss and symmetry-aware pooling mechanisms are introduced.

**Questions:**

Although tasks such as vaccine degradation is about mRNA sequences, currently most methods are using ncRNA models on them. Given that the proposed method converts the nucleotides to SO(2), is it possible to apply the proposed method on the tasks requiring single-nucleotide resolution?

**Ethical Concerns:**

["NO or VERY MINOR ethics concerns only"]

**Final Justification:**

> Why 5?

The proposed method applies group-theoretic principles, and achieves consistent improvements.

> Why not a higher score?

While the proposed method should definitely be accepted, but it is not yet award quality.

> Why not a lower score?

No major limitations are identified.

**Limitations:**

yes

**Paper Formatting Concerns:**

The manuscript is well-formatted.

**Quality:**

3

**Strengths And Weaknesses:**

## Strengths

### Novel and Principled Approach

The core idea of using group theory, specifically SO(2) rotations, to model the synonymous codon relationships is highly original and biologically principled. It moves beyond simply treating codons as unrelated tokens or using simple clustering losses, embedding a fundamental biological symmetry directly into the model's architecture. This is a significant conceptual contribution to genomic language modeling.

### Comprehensive Experimental Validation

The paper's empirical evaluation is thorough and convincing. The authors perform a detailed ablation study across 12 different model variants to dissect the contribution of each component (e.g., learned angles, Stiefel basis, equivariance loss). They test on a diverse set of six downstream tasks, covering expression, stability, and regulatory function.

## Weakness

### Lack of Comparison with Existing mRNA foundation models

Several foundation models pre-trained on mRNA sequences have been established in the field of RNA research. This includes [CaLM](https://huggingface.co/multimolecule/calm), [mRNA-FM](https://huggingface.co/multimolecule/mrnafm), and [AIDO.RNA cds variants](https://huggingface.co/multimolecule/aido.rna-1.6b-cds).
Comparing with these methods would significantly improve the validation of the proposed method.

Since a more detailed comparison with other methods is omission, I will not be able to grant a higher score.

---

> ### Author Rebuttal · Authors · 2025-07-31
>
> We thank the reviewer for their positive feedback and for recognizing the originality and biological motivation of our approach. We appreciate the acknowledgement of the theoretical contributions, the thorough ablation studies, and the broad applicability of Equi-mRNA across expression, stability, and regulation tasks. Below, we respond point-by-point to the specific concerns raised.
>  ## Weakness
>  `Lack of Comparison with Existing mRNA foundation models...`
>
> We have now implemented and evaluated CaLM [1], mRNA-FM [2], and AIDO.RNA [3] (cds variant) as baselines across the same downstream tasks. Each baseline was tuned using a comprehensive grid search, identical to the one used for Equi-mRNA, to ensure a fair comparison.
> We do note, however, that CaLM was not pre-trained on mRNA sequences, and AIDO.RNA was pre-trained primarily on non-coding RNA (ncRNA). As such, their application to coding sequence tasks may not fully reflect their intended use cases. Additionally, these models are 5–106× larger in parameter count compared to Equi-mRNA, which was deliberately designed to be lightweight and interpretable. Despite this, Equi-mRNA achieves competitive or superior performance, while offering biological interpretability and symmetry-aware modeling.
> Full results are included below. We believe these comparisons reinforce the practical and scientific value of our approach.
>
> **Table: Performance comparison across downstream benchmarks**
>
> | Model | E. coli(A) | MLOS(S) | iCodon(S) | Tc-Ribo.(S) | mRFP(S) | COV Deg(S) |
> |---|---|---|---|---|---|---|
> | **Nucleotide-Based** | | | | | | |
> | RNA-FM [2] (99.5 M) | 0.43 | - | 0.34 | 0.58 | 0.80 | 0.74 |
> | RNABERT [4] (82 M) | 0.39 | - | 0.16 | 0.47 | 0.40 | 0.64 |
> | Aido mRNA [3] (1.6B) | - | 0.461 ± 0.102   | 0.472 | 0.586  | 0.537  | *0.813* |
> | CaLM (86M) [1] | - | 0.430 ± 0.170 | 0.376 | 0.625  | 0.546| 0.773 |
> | mRNA-FM [2] (239M) | - | 0.509 ± 0.154  | 0.458 | 0.690 | 0.564  | 0.714  |
> | **Codon-Based** | | | | | | |
> | CodonBert [5] (82 M) | 0.57 | 0.543 | 0.350 | 0.502 | 0.832 | 0.78 |
> | GPT2 [6] (CLM)(50M)* | - | 0.611 | 0.498 | 0.531 | 0.815 | 0.787 |
> | GPT2 [6] (MLM)(50M)* | - | 0.653 | 0.503 | 0.569 | 0.753 | 0.801 |
> | HELM [6] (CLM)(50M)* | - | 0.592 | *0.529* | 0.619 | 0.849 | 0.789 |
> | HELM [6] (MLM)(50M)* | - | 0.701 | 0.525 | 0.626 | 0.822 | **0.833**|
> | **Equi-mRNA (5M)†** | *0.581* | *0.705 ± 0.12* | 0.519 | **0.764** | *0.853* | 0.756 |
> | **Equi-mRNA (15M)‡** | **0.613** | **0.710 ± 0.13** | **0.537** | *0.737* | **0.855** | 0.791 |
>
> † Mamba-based hybrid Architecture; ‡ GPT-2 Architecture both trained on 25M datapoints; *Trained on Antibody mRNA sequences
>
>
> ## Question
> `Although tasks such as vaccine degradation is about mRNA sequences...`
>
> We agree that certain tasks, such as vaccine degradation prediction, require single-nucleotide resolution, and that many existing models in this space are trained on non-coding RNA (ncRNA).
> Equi-mRNA is designed to model coding sequences (CDS) at the codon level, leveraging SO(2) rotations to capture the biological symmetry among synonymous codons. This structure is well-suited for translation-related tasks, but we acknowledge that it does not currently support per-nucleotide granularity, which is important for structure-sensitive or decay-related predictions.
> Adapting Equi-mRNA to handle such tasks would involve meaningful architectural changes. Specifically, it would require incorporating nucleotide-level embeddings and modifying the input resolution. As an initial direction, we are exploring a hybrid variant in which:
>  - Codon-level SO(2) embeddings are retained in the CDS;
>  - Nucleotide-level embeddings are used for UTR and regulatory regions;
>  - A shared transformer or attention-based architecture integrates both representations.
>
> This design would allow Equi-mRNA to maintain symmetry-aware modeling while extending its resolution and scope. Given the additional complexity, we have included this extension as part of our future work, and see it as a natural and exciting progression of the framework.
>
> ### References
> 1. Outeiral, Carlos, and Charlotte M. Deane. "Codon language embeddings provide strong signals for use in protein engineering." Nature Machine Intelligence 6.2 (2024): 170-179.
> 2. Chen, Jiayang, et al. "Interpretable RNA foundation model from unannotated data for highly accurate RNA structure and function predictions." arXiv preprint arXiv:2204.00300 (2022).
> 3. Zou, Shuxian, et al. "A large-scale foundation model for rna function and structure prediction." bioRxiv (2024): 2024-11.
> 4. Kalicki, Colin Hall, et al. "RNAbert: RNA family classification and secondary structure prediction with BERT pretrained on RNA sequences." 2020.
> 5. Li, Sizhen, et al. "Codonbert: Large language models for mrna design and optimization." bioRxiv (2023): 2023-09.
> 6. Yazdani-Jahromi, Mehdi, et al. "Helm: Hierarchical encoding for mrna language modeling." arXiv preprint arXiv:2410.12459 (2024).

---

> ### Comment · Reviewer_uj9Q · 2025-08-09
>
> Thank the authors for their rebuttal.
>
> My main concerns on lack of comparison with existing models in mRNA is addressed.
>
> Given the authors' rebuttal and the comments from other reviewers, I guess congratulations are in order.

---

### Comment · Area_Chair_CYfm · 2025-08-02
**Post-rebuttal Discussions and Finalizing Scores**

Dear Reviewers,

This paper receives 2 accepts and 2 borderline accepts before rebuttal. Please check if the authors have addressed your questions in the rebuttal and if you have any follow-up questions. Please read the author rebuttals, actively discuss with the authors before August 6, and finalize your scores before August 13.

Thank you for your contribution to the reviewing process!

Best regards,

Area Chair

---

### Decision · Program_Chairs · 2025-09-17

**Decision:**

Accept (poster)

**Comment:**

This work introduces Equi‑mRNA, the first codon‑level equivariant mRNA language model that explicitly encodes synonymous codon symmetries as cyclic subgroups of 2D Special Orthogonal matrix (SO(2)). The effectiveness is validated through extensive experiments. Reviewers recognize the merits of the paper including novel and principled method, as well as extensive experimental validation. The weaknesses include lack of comparison with existing mRNA foundation models, oversimplified assumptions for models, model scalability, experiment settings and results. The weaknesses and questions proposed by reviewers are addressed in the rebuttal by more experiments and clarifications, and all reviewers are satisfied with this work. Therefore, AC recommends accepting this paper.